# A Multi-Scale Study on Deformation and Failure Process of Metallic Structures in Extreme Environment

**DOI:** 10.3390/ijms232214437

**Published:** 2022-11-20

**Authors:** Zhi-Hui Li, Chenchen Lu, Aiqiang Shi, Sihan Zhao, Bingxian Ou, Ning Wei

**Affiliations:** 1China Aerodynamics Research and Development Center, Mianyang 621000, China; 2National Laboratory for Computational Fluid Dynamics, Beijing 100191, China; 3Jiangsu Key Laboratory of Advanced Food Manufacturing Equipment and Technology, Jiangnan University, Wuxi 214122, China; 4Special Equipment Safety Supervision Inspection Institute of Jiangsu Province, National Graphene Products Quality Inspection and Testing Center, Wuxi 214174, China

**Keywords:** molecular dynamics simulation, numerical simulation, mechanical response behavior, hypersonic reentry aerothermodynamic environment

## Abstract

It is a macro-micro model study for defect initiation, growth and crack propagation of metallic truss structure under high engine temperature and pressure conditions during the reentry atmosphere. Till now, the multi-scale simulation methods for these processes are still unclear. We explore the deformation and failure processes from macroscale to nanoscale using the Gas-Kinetic Unified Algorithm (GKUA) and all-atomic, molecular dynamic (MD) simulation method. The behaviors of the dislocations, defect evolution and crack propagation until failure for Aluminum-Magnesium (Al-Mg) alloy are considered with the different temperature background and strain fields. The results of distributions of temperature and strain field in the aerodynamic environment obtained by molecular dynamics simulations are in good agreement with those obtained from the macroscopic Boltzmann method. Compared to the tensile loading, the alloy structure is more sensitive to compression loading. The polycrystalline Al-Mg alloy has higher yield strength with a larger grain size. It is due to the translation of plastic deformation mode from grain boundary (GB) sliding to dislocation slip and the accumulation of dislocation line. Our findings have paved a new way to analyze and predict the metallic structural failure by micro-scale analysis under the aerodynamic thermal extreme environment of the reentry spacecraft on service expiration.

## 1. Introduction

Accurate prediction for the large-scale spacecraft’s flight track and landing area is still a great challenge during the reentering back to earth on service expiration. The truss structures sustain the structural response to deformation, softening, melting and disintegration with tremendous aerodynamic/thermal loading [1,2,3,4,5], due to the high-temperature thermochemical non-equilibrium gas flows. As preliminary research for the above issues, there is practical significance to develop the numerical forecast and analysis for predicting disintegration [3,6]. It reduces the risk posed to population and the property on the ground.

Macro-scale numerical simulation provides a reliable prediction of the spacecraft’s structural disintegration during the reentry process on service expiration. The structure deformation affects the temperature distribution in the interior of material significantly during the strong aerodynamic force and rigorous aerodynamic heating. A unified consideration and analysis to the coupled system of heat transfer, deformation and stresses on structure is needed to simulate the physical and mechanical problems [7]. Thus, the dynamic thermo-elasticity coupling response behaviors including material internal temperature distribution, structural deformation, and thermal damage are simulated by applying the Finite Element Algorithms (FEA) and GKUA in the field of hypersonic aerothermodynamics. Li et al. [5] study the structural deformation and thermodynamic response induced by reentry aerodynamic force and thermal loads. The large deformation and stress analysis of configuration was carried out by imposing mechanical and thermal loads [8]. Ji and Li et al. [9] proposed a structural dynamic model to study deformation of spacecraft under typical loads such as concentrated forces, body forces and torques. Studies showed that aerodynamic forces also have an important influence on nonlinear mechanical response of structures. The influence of thermal and mechanical loads should be considered at the same time. Furthermore, the GKUA [6,10,11,12] for solving the Boltzmann equations, combined with the direct simulation Monte Carlo (DSMC) method [13], Navier–Stokes/DSMC hybrid algorithm [14] and so on, has been developed. That method can be used to calculate the aerodynamic thermal, temperature, velocity distribution and other flow environments around the typical height of the aircraft accurately. However, the macro method cannot simulate the generation of atoms defects and the evolution of cracks on micro-scale. That micro behaviors play a crucial role in providing a reliable prediction for structural failure.

The MD simulation method can understand microstructural thermodynamics, physic properties and macroscopic transport properties at the atomic and molecular scale, which is widely used in physical chemistry, materials science, life science and other fields [15,16,17]. In the last decade, extensive research using MD simulations has been conducted to probe the mechanical behaviors of materials and microscopic mechanisms on structural failure. The generation and evolution of microstructure defects (voids, cracks, dislocations, twins and other defects) are dependent on temperature, loading patterns and material structure [18]. Numerous MD simulations based on atomic scale failure mechanism to the nanoscale void and crack were performed. The dynamic damage behaviors of crystal Zr reveal that the preference of void nucleation is dominated by temperature and strain rate [19]. Jing et al. [20] reports evolution mechanism of a spherical void in polycrystalline copper under uniaxial tensile loading, proposing a criterion that there exists a critical void diameter dependent on grain size. Lubarda et al. [21] pointed out the vacancy diffusion mechanism could be used only in the conditions of low strain rate and high temperature. They proposed a dislocation emission mechanism for void growth under high strain rate loading. The characterized void nucleation and final growth by the stress concentrations at a certain distance from the crack tip [22]. Furthermore, the structural gradient influences the stress distribution and the size of the plastic zone near the crack tip [23,24]. The grain size gradient can compromise the ability of the nano-grained metals to resist the crack propagation [25]. Gao et al. [26] indicated that the critical twin-boundary spacing for the onset of softening in nano-twinned copper and the maximum strength depends on the grain size by using MD simulation. MD method can estimate the mechanical properties and provide an understanding of the fracture mechanisms.

However, a systematical study on the simulation from the macro-scale numerical simulation to the MD simulation, which reveals the microscopic mechanism of failure of materials and predicts the disintegration of spacecraft metal truss structure in extreme aerodynamic thermal environment, is a concept that has rarely been studied.

In this work, a set of simulations are established for the degeneration and destruction of the metal (alloy) truss structure caused by the aerodynamic thermal extreme environment during the re-entry of the spacecraft. The gas dynamic theory unified algorithm of the Boltzmann model equation is used to provide the temperature and stresses distribution on the surface of the outflow field. Classical MD method is carried out at deformation and failure behaviors. We propose the mechanical properties and plastic transformation mechanism of Al-Mg Alloy at various temperatures, grain sizes, and strain rates. Macro-scale and micro-scale models and MD simulation provide a new analysis method for studying structural failure and damage caused by the aerodynamic thermal extreme environment.

## 2. Models and Methods

A set of macro and micro scale models are used to predict the structural failure of spacecraft truss. The truss structures sustain the structural response to deformation, softening, melting and disintegration with tremendous aerodynamic/thermal loading in Figure 1a. In microscale simulation, we employ a beam model to compute the temperature and pressure distribution at the truss structure surface in Figure 1b. Figure 1c shows the windward region of the model under a tensile loading. Figure 1d shows the leeward region of the model suffers a compression loading. The initiation and evolution of microstructure defects are dependent on temperatures, loading status and strain rates. With the consideration of the hypersonic reentry aerodynamic environment, we study the effect of temperature, loading pattern and strain rate on a truss structure.

The macroscale and microscale simulation details for the distribution of temperature and pressure are described in Section 3.1.1. In Section 3.2.1, a single crystalline Al-2%Mg model is applied to study the microstructure evolution of defects under different loading pattern and various temperatures. Moreover, a polycrystalline Al-2%Mg alloy model is carried out a simulation to study the mechanical properties. In Section 3.2.2, A pure Al model and a model with a central crack are used to study the defects generation and the propagation of cracks at different loading pattern. In Section 3.3.1, five polycrystalline Al-2%Mg alloy models are carried out a tensile simulation at various grain sizes to study the effect of strain rate.

In this study, we used the GKUA and FEA to obtain the temperature and pressure distribution of the outflow field. As a comparative verification of atomic simulation, an alloy cantilever beam model flying with a hypersonic velocity through a gas environment is simulated using the GPUMD (Graphics Processing Units Molecular Dynamics) [27]. The software LAMMPS [28] (Large-scale Atomic/Molecular Massively Parallel Simulator) is used to carry out the MD simulations for studying the mechanical properties and plastic mechanism. The embedded atom method (EAM) potential developed by Liu et al. [29] describes the interactions of Al-Mg alloy. Furthermore, the Voronoi method is used to model the polycrystalline alloy by the Atomsk software [30]. The dislocation extraction algorithm (DXA) [31] and Construct Surface Mesh [32] of the Open Visualization Tool (OVITO) [33] are utilized to reveal and analyze the variation in dislocation.

## 3. Results and Discussion

### 3.1. Aerodynamic Simulation

#### 3.1.1. Details of the Aerodynamic Model

A truss structure is simplified as a plate model with the size of 0.015 m × 0.5 m in macro aerodynamic simulation. We simulate the macro-mechanical behavior under the hypersonic reentry aerodynamic environment. The exterior of plate is the reentry near-continuous transition flow, the inflowing Mach number, Knudsen number and ratio of specific heat are Ma_∞_ = 8.37, Kn_∞_ = 0.01 and γ = 1.4, respectively. The exterior flow field of the plate with the mesh of 63 × 41 in the XOZ coordinate system is computed by the GKUA for solving the Boltzmann model equation with rotational non-equilibrium effect.

An atomic plate model is formulated with the size of 8.1 × 40.5 × 124 Å^3^ in nanoscale aerodynamic simulation. The size of MD simulation system box is 4050 × 40.5 × 1000 Å^3^ in the Figure 1b. The gaseous fill the system with 178,123 atoms. The initial states of the gaseous environment are set as pressure *P* = 30 atm and temperature *T* = 300 K. The mean free path of the gas is predicted to be 25 Å. The length of the beam is 124Å, and the corresponding global Knudsen number is 0.2. The lattice constant of the alloy atomic model is fixed at 4.05 Å. The velocity Verlet algorithm with a time step of 1 fs is used for the integration of the motion equations [34]. The plate moves along the negative X-direction with a speed of 1000 m/s in the canonical ensemble (NVT). The argon (Ar) atoms are coupled with Langevin thermostats at 300 K, and the alloy atoms are set as an adiabatic wall. The periodic boundary condition is applied to the three directions. The system is carried out for at least 6 ns to guarantee the system reaching a steady state. The atom interaction can be accurately modeled with the 12-6 Lennard-Jones (LJ) potential [35] for Ar-Ar, Ar-Al and Ar-Mg.
(1)$$U_{\left(r_{ij}\right)}=4\varepsilon \left({\left(\frac{\sigma }{r_{ij}}\right)}^{12}-{\left(\frac{\sigma }{r_{ij}}\right)}^{6}\right)$$
where *ε* and *σ* are the depth of the potential well and the distance at which the LJ interaction is zero, and *r_ij_* is the distance between two particles. For Ar-Ar interaction, *ε* = 0.010325 eV and *σ* = 3.828 Å. For Ar-Mg interaction, *ε* = 0.0062141 eV and *σ* = 3.0687 Å. For Ar-Al interaction, an LJ potential with parameters *ε* = 0.01325 eV and *σ* = 3.7271 Å [36] is used as a reference potential. The LJ potential cutoff radius is set as 10 Å.

The EAM potential [37] is used to described Al-Al, Al-Mg and Mg-Mg interactions of Al-Mg alloys.
(2)$$U_{i}=U_{\alpha }\left(\sum_{j\neq i}\rho_{\beta }\left(r_{ij}\right)\right)+\frac{1}{2}\sum_{j\neq i}\phi_{\alpha \beta }\left(r_{ij}\right)$$
where *r_ij_* is the separation distance between atoms *i* to *j*, *U_α_* is the energy required to embed an atom in the electron cloud, and *ρ*(*r_ij_*) is the electron transfer function between the atoms. The pairwise part *Φ*(*r_ij_*) is the interaction of the atom *i* to *j*. The desired Al-Mg alloys are modeled by replacing a selected percentage of Al atoms with the same ratio of Mg atoms at the same locations, as shown in Figure 1c,d.

#### 3.1.2. Outflow Field Analysis of a Plate in Hypersonic Reentry Flow

The outflow field obtained from macro numerical simulation and micro atomic simulation is shown in Figure 2a–e. The temperature and pressure around plate from the GKUA are shown in Figure 2a,b. Because of the reentry near-continuum transition flow region, the thick detached shock wave layer is formed around the thin plate. The strong aerodynamic heating and force are imposed as the interface condition of the present thermodynamic on the plate surface. The incoming hypersonic flow crosses the strongly disturbed detached shock wave layer in front of the stagnation points region, reaching the highest temperature and stress. The high temperature and strong pressure make the plate expand and deform. The airflow expands rapidly into the leeward region. The temperature and pressure decrease to the lowest at the rear stagnation point. Therefore, we can obtain the actual temperature *T_a_* and *P_a_* values conducted to the plate boundary through the following formula [5]:(3)$$T_{a}=T\cdot T_{\infty }$$
(4)$$P_{a}=P\cdot \left(\frac{1}{2}\rho_{\infty }V_{\infty }^{2}\right)$$
where *T_∞_*, *ρ_∞_* and *V_∞_* represent the reference temperature, gas density and velocity of the external flow field, respectively. *T* and *P* represent the temperature and pressure parameters of the external flow field calculated by GKUA, respectively. The *T* and *P* in the right side of Equations (3) and (4) are dimensionless components.

The number density, temperature and pressure are solved by MD simulation in Figure 2c–e. microscopic observables, including temperature *T* and pressure *P* are calculated as the spatial and time average in the volume of a prescribed grid cell by the following formula [38]:(5)$$T=\frac{2}{3Nk}\sum_{a}^{N}\frac{m_{a}v_{a}^{2}}{2}$$
(6)$$P_{ij}=\frac{\sum_{a}^{N}m_{a}v_{ai}v_{aj}}{V}+\frac{\sum_{a}^{N}r_{ai}\cdot F_{aj}}{V}$$
where *T* is the temperature, *k* is the Boltzmann constant, *N* is the number of atoms, *V* is the volume of grid cell, *m_a_* is the mass, ***v****_a_* is the velocity vector, ***r****_a_* is the position vector, and ***F****_a_* is the force vector of the atom *a*. ***P****_ij_* is the pressure tensor (where *i* and *j* are x, y, or z). The volume of the grid cell with a size defined as Δx × Δy × Δz. When drawing the outflow distribution in the XOZ plane, the size of Δx × Δz is set as 1 Å × 1 Å for density and temperature, and 2 Å × 2 Å for pressure. The length of Δy is set as the length of the box in the Y-direction.

The steady results are given by the MD simulation and the number density, temperature, and pressure distribution are obtained in Figure 2c–e. Figure 2c demonstrates the number density distribution, and the gas atoms are mainly concentrated on the surface area of the plate. The maximum density appears in the leeside of the plate in the bottom. It is similar to the pressure distribution in Figure 2e. The compression of the gas leads to the increase in gas density. Figure 2d shows the external temperature distribution of flow field. The maximum temperature of the freestream is located in the shock wave area in front of the surface of the plate. From the external plate-surface temperature and pressure distribution of flow fields, the increase in surface temperature and pressure (compared to the inflow values) of the windward area for the left side flat plate is larger than those of the leeward area for the right-side flat plate.

Compared with macro and micro simulations, the flow field results are in good agreement. The windward boundary temperature of the plate rises gradually along the positive Z-direction in Figure 2d. Over the top surface, the temperature drops rapidly to freestream temperature. The temperature reaches the maximum value in X-direction near the windward stagnation point. It is evident that it has the maximum curvature variation and is subjected to the most severe aerothermodynamic heating. From Figure 2e, the maximum pressure appears at the left bottom region. The gas atoms gather in this area, showing the maximum atomic density in Figure 2c. Increasing atomic density leads to the increase in pressure due to the compression of gas. Meanwhile, the results of temperature and pressure distribution system obtained from the macroscale and microscale models are well consistent with each other. It verifies the feasibility and validity of this macro-scale and micro-scale simulation methods under extreme environment.

Between the macroscopic and microscopic models, we adopted the simplified model, and obtained the temperature and pressure distribution on its surface through the macroscopic model and the microscopic system. We obtained the similar phenomenon. The distribution and variation of temperature and pressure are in good agreement with macroscale and microscale results. As shown in Figure 2, the incoming hypersonic flow crosses the disturbed detached shock wave layer in front of the stagnation points region, reaching the maximum temperature and pressure around the plate. The airflow expands into the leeward region rapidly. The temperature and pressure decrease to the lowest value at the rear stagnation point. The highest temperature value appears at the left top inflection point of the plate. It has the maximum curvature variation and is subjected to the most severe aerothermodynamic heating. The maximum pressure appears at the bottom of the plate due to the increasing atomic density leads to the compression of gas. The consistency of macro and micro models is confirmed.

We calculate the temperature and pressure distribution on the plate surface quantitatively by MD simulation in Figure 3. When plotting the curves of microscopic parameters along the Z-direction, we set the length of Δz as 10 Å and the size of Δx × Δy is 20 Å × 40.5 Å. We calculated as the spatial and time average in the volume of a prescribed grid cell for every atom temperature and virial stress. Due to the hypersonic reentry flow around, the plate is subjected to aerodynamic heating and force. Form the Figure 3, the pressure on the surface of the windward boundary decreases along with positive Z-direction. Over the top surface, the pressure drops to 0 Pa. On the one hand, the compression of gas causes the increase in atom density in the left bottom surface. On the other hand, the airflow expands rapidly into the leeward region after through the corner of the plate, resulting in a decrease in pressure. The maximum pressure and the temperature values appear at the bottom and the top of the beam, respectively. The maximum pressure is approximately 490 Mpa, and the highest temperature is about 680 K.

The distribution of temperature and stress provide thermo-mechanical states of metallic structural states under different mechanical and thermal coupling conditions. It shows the windward region under the tensile loading and the leeward region under the compressive loading, respectively. In addition, the boundary conditions of the distribution of temperature and stress are used in the microscale model. With the consideration of the hypersonic reentry aerodynamic environment, we study the effect of temperature, loading pattern and strain rate on truss structure.

### 3.2. Deformation and Failure of Metal Model in Extreme Environment

#### 3.2.1. Model and Validation

The desired Al-Mg alloys are modeled by replacing a selected percentage of Al atoms with the same ratio of Mg atoms at the same locations [38] in Figure 4a. For the single crystal model, its dimensions are 48 × 48 × 48 Å^3^, while they are 405 × 405 × 49 Å^3^ for the polycrystalline model. The single crystal orientation of [100], [010] and [001] is used in X, Y, and Z directions, respectively. An unit Al model is established using the Atomsk software [30]. The volume size, grain number and grain orientation are set to generate polycrystalline Al model. We delete the atoms whose distance space is less than half the lattice constant [39]. The polycrystalline models of Al-Mg alloy with 2% Mg content are formed by substituting 2% Al atoms with Mg atoms. The atomic model with grain size d = 12.8 nm is shown in Figure 4b.

The periodic boundary conditions are applied in all three-dimensional directions to avoid surface effects. The samples are relaxed by minimizing the energy using the conjugate gradient algorithm, where the force and energy tolerances are 10^−12^. The samples are performed to relaxation (60 ps) in the NPT ensemble (Nosé-Hoover isobaric-isothermal ensemble) with a timestep of 0.001 ps. The structures are optimized when the system potential energy converges. During the tensile deformation simulation, the system temperature is controlled by rescaling the velocity of atoms. Subsequently, a uniaxial tensile loading is carried out (*Z*-axis) with a strain rate of 10^9^ s^−1^. Uniform uniaxial tension with constant strain rate was modeled by the scaling of coordinates of atoms with using the ‘deform’ command of LAMMPS. The uniaxial tensile simulation is to squeeze the material throughout its length, which leads to elongation along the Z-direction. This can also be visualized as a change in the simulation box size to deform the material. The interaction of dislocation, with other dislocations, GB, and precipitates are usually studied using this method.

The simulations on the defect-free pure Al models are carried out to verify the reliability of the calculation results. The results are shown in Table 1. The Young’s modulus is obtained by linear fitting of strain ε = 0.6–1.4% [40] in the elastic stage of the stress-strain curve. The point of maximum stress is considered as the yield stress during the tensile simulation. The results are well consistent with the study by Zhu et al. [40], and the reliability of simulation results is verified.

#### 3.2.2. The Model of Pure Al and Model with Central Crack

Cracks cause catastrophic failure of structural materials. The information about their initiation and propagation within the grain is important. MD can be used to study the crack configuration at the nanoscale. A pure Al model and a model with a central crack are used to study the mechanical properties, respectively. As shown in Figure 5a, the dimensions of the perfect Al are approximately 202.5 × 202.5 × 24.3 Å^3^, which is modeled as a regular face-centered cubic (FCC) lattice with the initial orientation of [100], [010], and [001] in the X, Y, and Z directions, respectively. In the model with a center crack, as shown in Figure 5b, the length of the initial crack a is 2.83 nm, and the width b is 0.81 nm. The models are conducted to relaxation (200 ps) using the NPT ensemble (Nosé-Hoover isobaric-isothermal ensemble) at 300 K and 0 bar with a timestep of 0.001 ps. The uniaxial tensile loading is carried out (Y-direction) with a constant strain rate of 0.0005 ps^−1^. In compression simulation, the model is compressed at a strain rate of 0.0005 ps^−1^ in the Y-direction until the strain reaches 12%.

#### 3.2.3. The Deformation Behaviors of Pure Al under Tension and Compression Loading

This section focused on the pure Al mechanical response and failure microscopic mechanism under uniaxial tension and uniaxial compression. Figure 6 shows the stress-strain curve and dislocation analysis under different loading patterns. The yield stress under tension and compression are 7.51 GPa and 5.77 GPa, respectively. The inset figure in Figure 6a shows the dislocation analysis at the yield point and failure point under uniaxial tensile loading. The inset in Figure 6b shows the dislocation analysis at the yield point and failure point under uniaxial compressive loading. At yield point, the dislocations emit from internal body and lead to the stacking faults (SFs), as shown in Figure 6a. Shockley partial dislocations dominate the generation of defects. In plastic deformation stage, the dislocation types are dominated by perfect dislocation and Shockley partial dislocation. The dislocation lines are generated around the hexagonal close packed (HCP) structure atoms. From the dislocation analysis in Figure 6b, dislocations emitted mean that the system has entered the plastic deformation stage. The Shockley dislocations and the Stair-rod dislocations dominate the plastic deformation at compression loading. The entanglement of SFs generates the stair-rod dislocations between two slip surfaces, as shown the atomic configuration in a black circle.

The statistics analysis of HCP atoms and dislocation lines at two loading patterns are illustrated in Figure 7. The curves of HCP atom percentage and dislocation density with the strain at tensile loading are monitored in Figure 7a,b, while the curves at compression loading is presented in Figure 7c,d. A similar result can be found between tensile and compressive loading. The ratio of HCP atom increases sharply in the plastic stage, and declines with the increase in strain after reaching the peak value. In the meantime, the peak points of the HCP atom ratio and dislocation density appeared at a same strain. Obvious peak values can be observed from Figure 7 at different loading. A peak value of the ratio HCP atom at compression condition is found around 9%. This is almost 9 times greater than the value at tensile process. However, the compressive process obtains a smaller peak value of dislocation density.

Figure 8 represents the curve of the crack length with strain in the pure Al model and the shape of the crack propagation under tensile loading. The growth process of crack is roughly subdivided into three different stages from Figure 8a. An elastic deformation is shown in stage I. A stable crack length is observed in this stage. There is no dislocation around crack in Figure 8b, indicated a stable lattice structure. In stage Ⅱ, the material enters the plastic deformation stage. The length of crack increases from 2.74 nm to 7.23 nm sharply when the strain increases. The maximum central symmetry parameter (CSP) value is observed near the crack tip, meaning the generation of initial defects. The inset shows the atomic structure around crack by Common Neighbor Analysis (CNA). From Figure 8c, defects are consisted of SFs and disorder atoms, resulting in the plastic deformation (as shown in the inset figure, where red atoms represent the SFs, green atoms represent the FCC structure, and white atoms represent the disordered structure). In stage III, the crack grows slowly. From Figure 8d, disordered structures around crack tip inhibit further crack growth thus increasing the critical stress for crack propagation. Defects are distributed from crack tip to the whole system.

#### 3.2.4. The Evolution of Atomic Defects in Al-2%Mg Single Crystal Alloy with Different Temperatures

Tensile simulations of the mechanical properties and failure mechanism of Al-2%Mg single crystal alloy are carried out at temperatures ranging from 10 K to 500 K. Stress-strain curves of Al-2%Mg alloys at different temperatures are plotted in Figure 9. The four curves show a linear relationship before reaching the yield point. Four points, a_1_, b_1_, c_1_ and d_1_, are the maximum stress at 10 K, 100 K, 300 K and 500 K, respectively. After the point of maximum stress, the stress drops simultaneously as the strain increases, showing the generation and growth of defects, which was considered as failure in the works. As the strain increases, the samples enter the plastic deformation stage. From Figure 9, the effect of temperature on yield stress is significant. At 10–500 K, the yield stress decreases sharply from 9.12 Gpa to 5.43 Gpa. As the temperature increases, the yield stress decreases due to more phonons absorbed by lattice, the increased atomic mobility and the rapid diffusion of free volume. Hence, the atoms can be displaced more easily at higher temperature when the external loading is applied.

Uniaxial tensile loading is carried out to investigate the evolution mechanism of microstructure for Al-2%Mg single crystal alloy under different temperatures. In Figure 10, the atomic structure snapshots present the defects evolution at every 0.01% strain from the yield point under four temperatures condition (10 K, 100 K, 300 K and 500 K). Corresponding to the stress-strain curves in Figure 9, the snapshots appear initial dislocation at the strain of the yield points. As the temperature increases, the strain of initial dislocation emission decreases obviously. In addition, we analyze the initial dislocation types at different temperature, Shockley dislocations are observed in all samples. The dislocations begin to grow and slide when the strain increases. At high temperature, the strength decreases depending on the less defective structures and short dislocation lines. Meanwhile, higher proportion of HCP atoms strengthens the structure stability at low temperature.

Uniaxial compression simulations of the mechanical properties and failure mechanism of Al-2%Mg single crystal alloy are carried out at temperatures ranging from 10 K to 500 K. Stress-strain curves of Al-2%Mg alloys at different temperatures are plotted in Figure 11. Four curves show a nonlinear relationship before reaching the yield point. Four points, a1, b1, c1 and d1, are the maximum stress at 10 K, 100 K, 300 K and 500 K, respectively. As the increase in strain, the samples enter the plastic deformation stage, and the stress drops sharply. The increase in temperature decreases the yield strength and elastic modulus, which is consistent with the phenomenon in tensile simulations. At the end of elastic deformation stage, the stress increases slowly as the configuration is strained, which is different from the stress-strain curves in tensile process.

Uniaxial compression loading was carried out to investigate the evolution mechanism of microstructure for Al-2%Mg single crystal alloy under different temperatures. In Figure 12, the atomic structure snapshots present the defects evolution at every 0.01% strain from the yield point under four temperatures condition (10 K, 100 K, 300 K and 500 K). Corresponding to the stress-strain curves in Figure 11, the snapshots appear initial dislocation at the strain of the yield points. As the temperature increases, the strain of initial dislocation emission decreases obviously. The dislocations begin to grow and slide when the strain increases. Different from tensile processes, the proportion of HCP atoms decreases not obviously with the increase in the temperature at compression loading pattern. Moreover, more defective structures are generated at compression loading condition. In the process of compression simulation, the damage to the lattice structure is more intense.

The process of crack propagation is investigated at different conducted conditions. The Shockley partial dislocation dominates the deformation in tensile process. The stair-rod dislocation dominates the deformation in the compression process. Furthermore, the simulations are carried out at various temperatures to study the mechanical properties. The increasing temperature decreases the strength under different loading patterns. Meanwhile, the tensile loading causes less defective structure comparing to compressive loading. The structure evolution of defects indicates that the alloy strength is more sensitive to compressive loading. During the hypersonic reentry aerodynamic environment, the windward and leeward regions suffer different temperature and loading condition. The effect of temperature and loading pattern are systematically analyzed by MD simulation. Those results can provide an atomistic-scale mechanism on the structure failure by defects evolution.

### 3.3. The Mechanical Properties and Plastic Transformation Mechanism in Al-2%Mg Alloy

#### 3.3.1. Effect of Grain Size on Mechanical Properties and Structural Evolution of Al-2%Mg

The nanocrystalline Al-2%Mg alloys models are carried out a tensile simulation to study the mechanical properties with average grain sizes of 5.1 nm, 7.0 nm, 10.1 nm, 12.8 nm and 16.0 nm, respectively. In the MD simulations with periodic boundary conditions, the mechanical properties of defect-free metals have no effect on size [42]. The detailed parameters of these models are shown in Table 2.

Uniaxial tension simulations are used to study the mechanical properties for nanocrystalline Al-2%Mg alloy with various grain size. According to the stress-strain curve in Figure 13a, a linear relationship is appeared at the elastic stage. The peak point of the stress-strain curve is defined as the yield stress. After the point of maximum stress, the stress drops sharply as the strain increase, indicating the samples enter the plastic stage. The stress rises again, showing a repeated yield phenomenon. It is similar to the zigzag stress-strain curve reported by Tong et al. [43] in the tensile experiment of Al-Mg alloy. This phenomenon was also observed in the simulation study of mechanical properties of zirconium nanowires by Guder et al. [44]. In Figure 13a, the initial stress value is higher than zero at small grain size. With the increase in grain size, the stress approaches to zero. The reason is that the large proportion of atoms locating at GB leads to a high internal stress after relaxation. This result is consistent with Wang et al. [45] in studying the mechanical properties of nickel metals.

We further explore the mechanical properties under different grain size. The yield strength of each sample with various grain sizes is plotted in Figure 13b. The peak point of the stress-strain curve is defined as the yield stress. The elastic modulus is obtained by linear fitting of strain ε = 0.6–1.4% [46] in the elastic stage of the stress-strain curve. The yield strength increases with the increase in grain size. In nanocrystalline samples, the proportion of GB atoms influences the mechanical properties obviously. A small grain size means a large proportion of GB atoms in model. Based on the analysis of the GB atoms ratio at various grain size model, the GB atoms ratio is as high as 19.2% while the grain size is 5.1 nm. At large grain size model with 16.0 nm, the GB atoms ratio decreases sharply by 67.7%. The GB atoms ratio is only 6.2%. The numerical results indicate that the increase in grain size decreases the initial defects caused by dislocations at GB. The yield strength is improved at large grain size.

The effect of grain size on the plastic deformation of Al-2%Mg alloy is further studied at various grain sizes. The dislocation density of samples with five grain sizes (5.1 nm, 7.0 nm, 10.1 nm, 12.8 nm and 16.0 nm) is plotted in Figure 14. The increase in grain size decreases the dislocation density at the same tensile strain. The reason is that the increase in grain size decreases the proportion of GB atoms. 

The dislocation density decrease as the strain increases at the grain size of 5.1 nm. In the small grain size, the GB sliding is observed at plastic deformation stage, following few grain deformations. In the movement of GB, the interaction of GB sliding, and grain rotation releases the dislocation lines, which is generated by lattice mismatch at the GB. The dislocation generated by grain deformation have small proportion of total dislocation density. The dislocation density decreases with the increase in grain size obviously. The plastic deformation increases the dislocation density slightly of the sample with the grain size of 16.0 nm. Compared with the initial dislocation density, a variation in plastic deformation mode is observed from above results. The plastic deformation is dominated by grain boundary sliding in the small grain size, while it is dominated by dislocation slip in the large grain size. 

The atomic structure evolution snapshots of the samples with different strains are plotted to explore the plastic deformation. From Figure 15a,b, the lattice structure has no noticeable variation in the elastic deformation stage, which corresponds to the linear growth stage of the stress-strain curve. With the increase in loading, the thickness of GB broadens gradually, meaning the sample enters the plastic deformation stage. At the same time, a few dislocations nucleate at the GB, as shown in Figure 15c. The new dislocations are emitted from the GB and moved to grains interior, driven by stress. Most dislocations are hindered and absorbed by the GB. In Figure 15d, a few dislocations move through the GB to the adjacent grains. In the plastic deformation stage with small grain size, the large proportion of the GB atoms dominate the plastic deformation. Compared with the initial structure, the grain has been variated by the GB movement, as shown in Figure 15e,f. In the GB movement, the GB slip contributes to the variation of grain shape. From Figure 15f, the grain distorts with the increase in GB width at tensile process. The plastic deformation of the sample with grain size d = 5.1 nm is dominated by the movement of GB and propagation of a few dislocations.

The atomic configuration of grain size d = 16.0 nm under different strains is shown in Figure 16. The sample stays in the elastic stage until reaching the yield point in Figure 16a,b. Compared with the atomic configurations of grain size d = 5.1 nm in Figure 15, more dislocations are generated at sample with large grain size. From Figure 16d, the dislocation slip contributes to the plastic deformation. The shape of grains is changed obviously during the plastic deformation process. A large grain size decreases the proportion of GB, resulting in the plastic deformation is dominated by dislocation slip. With the further increase in the grain size, a few defects are observed to nucleate in the grain during the plastic deformation process with the grain size of d = 16.0 nm. A few atoms in the grain appear local amorphization (yellow atoms in the grain), as shown in Figure 16c. Compared with the atomic structure of grain size d = 5.1 nm, more dislocations nucleation is observed at GB, as shown in Figure 16d,e. In the plastic deformation stage, the structure of each grain remains intact, as seen in Figure 16f. In the sample with large grain size, the dislocation nucleation dominates the plastic deformation.

#### 3.3.2. Effect of Strain Rates on Plastic Mechanism of Polycrystalline Al-2%Mg Alloy

The strain rate is related to nanocrystalline mechanical properties and plastic mechanisms. The microstructure deformation mechanism of the materials has different behaviors at varied strain rates. From Section 3.3.1, the plastic deformation mechanism depends on the grain size. The plastic deformation is dominated by GB movement in the small grain size, while it is dominated by dislocation slip in the large grain size. Therefore, to explore the strain rate dependent plastic deformation behavior of the nanocrystalline Al-2%Mg alloys with varied grain sizes, the samples with grain sizes (7.0 nm and 16.0 nm) are subjected to uniaxial tensile loading at 300 K. The strain rates used for tensile simulation are 5 × 10^8^ s^−1^, 10^9^ s^−1^, 5 × 10^9^ s^−1^, and 10^10^ s^−1^, respectively.

The dislocation density of the samples at various strain rate loading is plotted in Figure 17. The trends of dislocation density with strain rate are consistent in four samples until the strain reaches 6%. The strain rate does not influence the elastic deformation process of the nanocrystalline Al-2%Mg alloy. However, the higher strain rate leads to larger dislocation density with the increase in strain at plastic deformation stage. It is indicated that increasing the strain rate is conducive to dislocation nucleation and propagation, which shows a similar trend in the samples of various grain sizes. Compared to small grain size, the samples with large grain size obtain a large dislocation density at the same strain, as shown in Figure 17a,b.

The atomic configurations with grain size d = 7.0 nm and d = 16.0 nm at different strain rates are plotted to explore the effect of strain rate at the plastic deformation stage, as shown in Figure 18. From Figure 18a–d, with the increase in strain rate, dislocations increase significantly in grain size d = 7.0 nm. The atomic configuration gets disordered. Increasing strain rate leads to severe plastic deformation of the GB. As for the system with grain size d = 16.0 nm in Figure 18e–h, no noticeable movement of GB is observed except for the increase in dislocation nucleation. In Figure 18g,h, the grains are filled with SFs with increasing the strain rate, which is well consistent with the increase in dislocation density in Figure 17b. Hence, increasing strain rate is conducive to dislocation propagation. The degree of the dislocation propagation is dependent on grain size. The GB movement dominates the plastic deformation in small grain size. The dislocation slip dominates the plastic deformation in large grain size. A high strain rate increases the density of dislocations in plastic deformation stage. During the deformation process, a few dislocations nucleate from the grain internal and entangle with the dislocations emitted from the GB, resulting in severe dislocation stacking. In Figure 18e,f, unchanged GB configuration is observed clearly. Compared with initial system, the GB configuration with large grain sizes is insensitive to the strain rate.

Figure 19 depicts the dislocation lines with grain size d = 16.0 nm under different strain rates, where white structure and lines segments represent the GB and dislocation lines, respectively. The system has few dislocation lines at a strain rate of 5×10^8^ s^−1^. As the strain rate increases, the dislocation lines increase remarkably, and the system is filled with dislocation lines when the strain rate is up to 10^10^ s^−1^. The increased strain rate leads to the propagation and tangles of dislocation lines in the samples. Meanwhile, the above analytical results are verified. The sample has higher yield stress with the strain rate increased.

For polycrystalline Al-Mg alloy model, the yield strength increases with the increasing grain size. The proportion of GB atoms influences the dislocation density. The generation of dislocation decreases the yield strength in small grain size. The plastic deformation is dominated by grain boundary sliding in the small grain size, while it is dominated by dislocation slip in the large grain size. The nucleation and propagation of dislocation increase the yield strength at higher strain rate loading process. In addition, the strain rate corresponds to the boundary conditions of the distribution of stress on the spacecraft surface. Large pressure on the surface means a high strain rate of samples. Those results can understand the mechanism of structure failure under hyperthermal pressure condition.

## 4. Conclusions

In summary, the metallic deformation and failure process is investigated systematically via macro-micro simulations (GKUA and MD) methods. The results obtained from the MD simulations are well consistent with those from the GKUA method. The crack propagation can be separated into three stages for a pristine aluminum model with an initial crack, according to the relationship between crack length and strain. The yield strength of Al-Mg alloy decreases sharply with the increase in temperature, while it is enhanced with the increasing grain size of polycrystalline alloy. It’s attributed to the plastic deformation mode dominated by GB sliding in the smaller grain size to the plastic deformation dominated by dislocation slip in the larger grain size. Our findings hold great promise to inspire a fresh perspective in providing micro-scale analysis method to predict the metallic structural failure behaviors under the aerodynamic thermal extreme environment of the reentry spacecraft on service expiration.

## Figures and Tables

**Figure 1 ijms-23-14437-f001:**
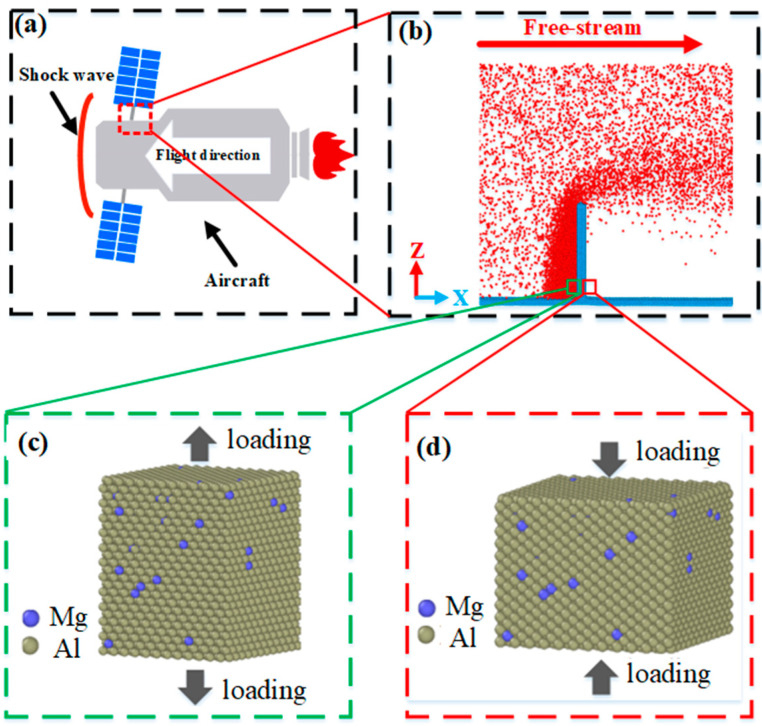
Molecular dynamics model of cantilever beam. (**a**) The structural failure of spacecraft truss structure. (**b**) The simplified truss model for MD simulation. (**c**) The windward region of the model suffers the tensile loading. (**d**) The leeward region of the model suffers the compression loading. The black arrows denote the tensile loading in (**c**) and compression loading in (**d**).

**Figure 2 ijms-23-14437-f002:**
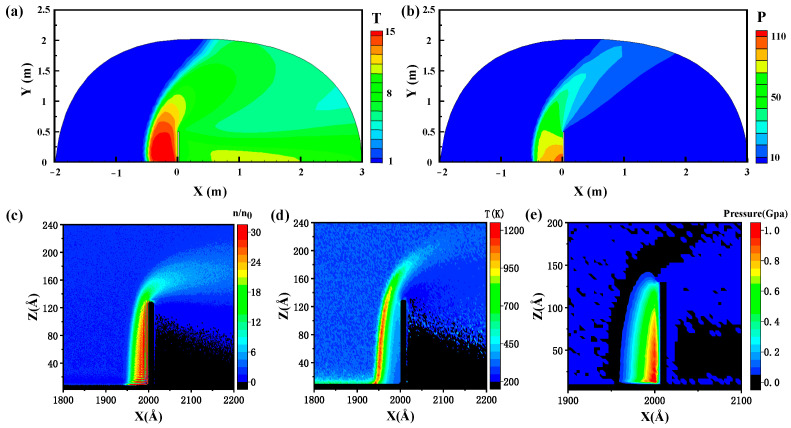
Reentry hypersonic flow field (Ma_∞_ = 8.37, Kn_∞_ = 0.01) of vertical plate in near-continuum transition flow solved by GKUA in (**a**,**b**). (**a**) Temperature distribution of outflow field; (**b**) Pressure distribution of outflow field. *T* and *P* translate to actual temperature and pressure values by Formulas (2) and (3). Reentry hypersonic flow field (Ma_∞_ = 3.2, Kn_∞_ = 0.2) of vertical plate in near-continuum transition flow solved by MD simulation in (**c**–**e**). (**c**) Number density distribution of outflow field; (**d**) Temperature distribution of outflow field; (**e**) Pressure distribution of outflow field.

**Figure 3 ijms-23-14437-f003:**
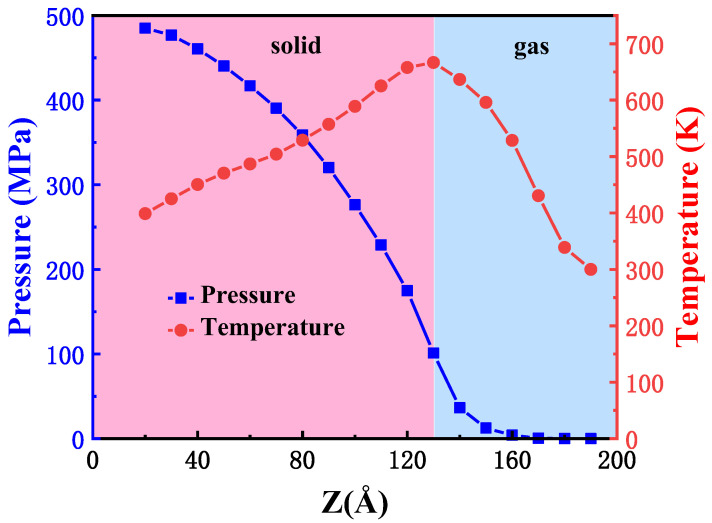
Pressure and temperature distribution along the plate surface solved by MD simulation. The blue square denotes pressure, and the red circle denotes temperature.

**Figure 4 ijms-23-14437-f004:**
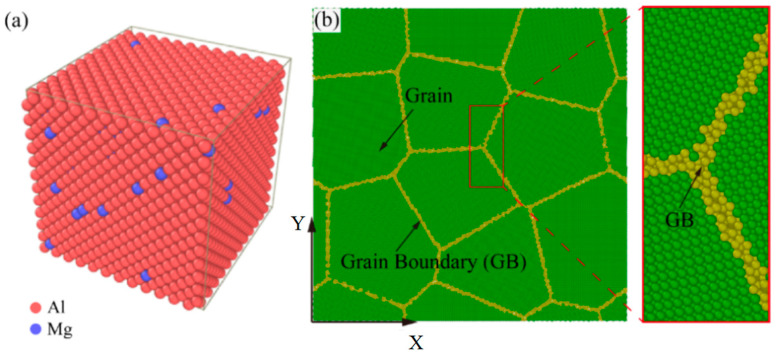
Molecular dynamics simulation models for Al-Mg alloys with 2% Mg content. (**a**) The single crystal Al-Mg alloy model. The red atoms denote the aluminum atom, and the blue atoms denote magnesium atom. (**b**) The polycrystalline Al-Mg alloy model with grain size d = 12.8 nm. The green atoms represent grain, and the yellow atoms represent GB.

**Figure 5 ijms-23-14437-f005:**
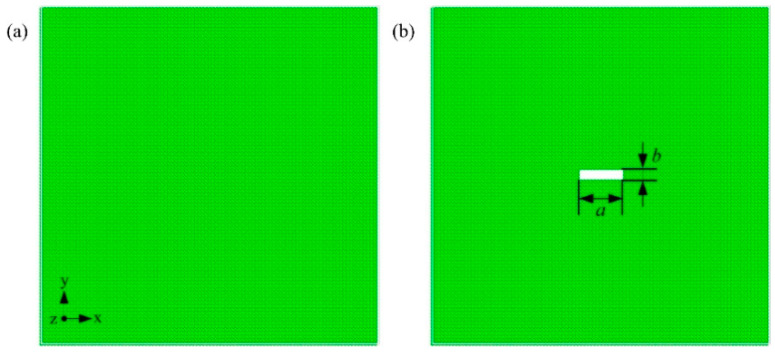
(**a**) The pure aluminum model with the size of 20.25 × 20.25 × 2.43 nm^3^; (**b**) The model with central crack. The length of the initial crack a = 2.83 nm, and the width b = 0.81 nm.

**Figure 6 ijms-23-14437-f006:**
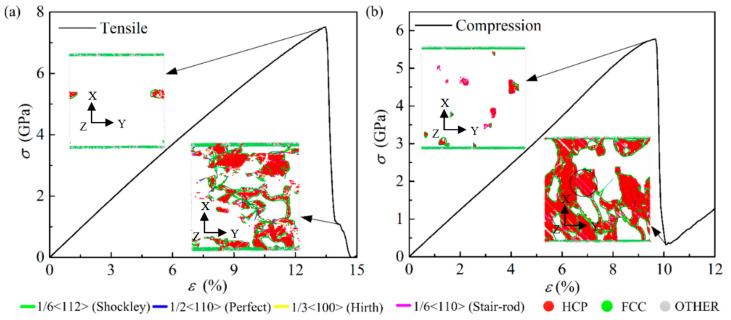
Stress-strain curve and dislocation analysis of pure Al at different loading patterns. (**a**) Uniaxial tension and (**b**) uniaxial compression. The insets figures present the atomic structure by common neighbor analysis for the dislocation emitted under tension and compression loading. The green and red atoms represent FCC and HCP atoms, respectively.

**Figure 7 ijms-23-14437-f007:**
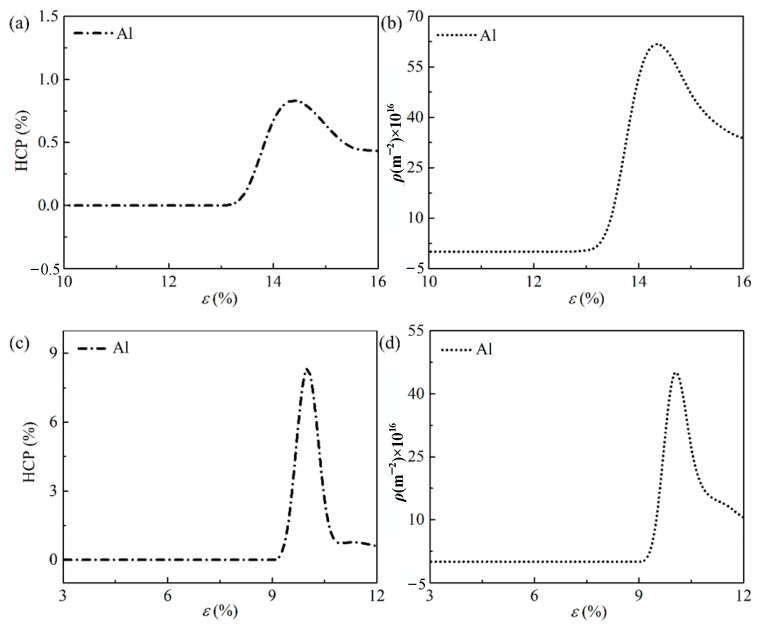
HCP atom percentage and dislocation density analysis of pure aluminum under different loading. (**a**,**b**) Tensile loading; (**c**,**d**) Compression loading.

**Figure 8 ijms-23-14437-f008:**
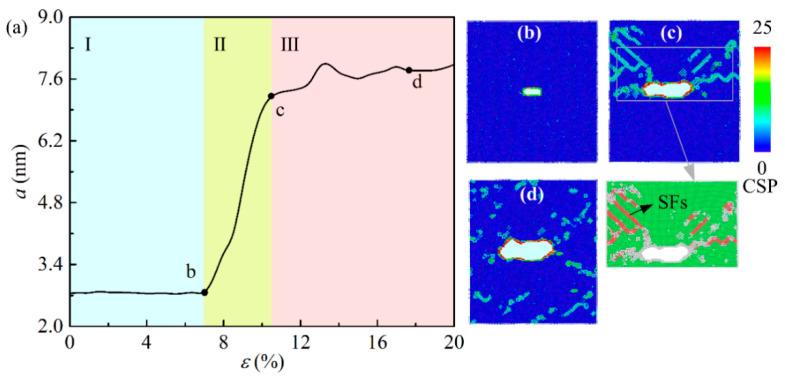
The relationship between crack length and strain curve under uniaxial tensile. (**b**–**d**) corresponds to the CSP atomic configuration diagram of the feature point in (**a**).

**Figure 9 ijms-23-14437-f009:**
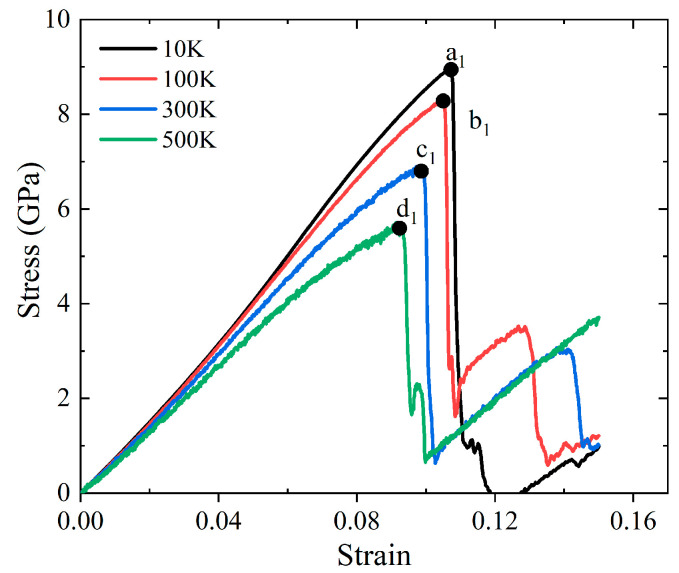
Stress-strain curves for Al-2%Mg alloys under uniaxial tension with different temperature. The four points, a_1_, b_1_, c_1_ and d_1_, are the maximum stress at 10 K, 100 K, 300 K and 500 K, respectively.

**Figure 10 ijms-23-14437-f010:**
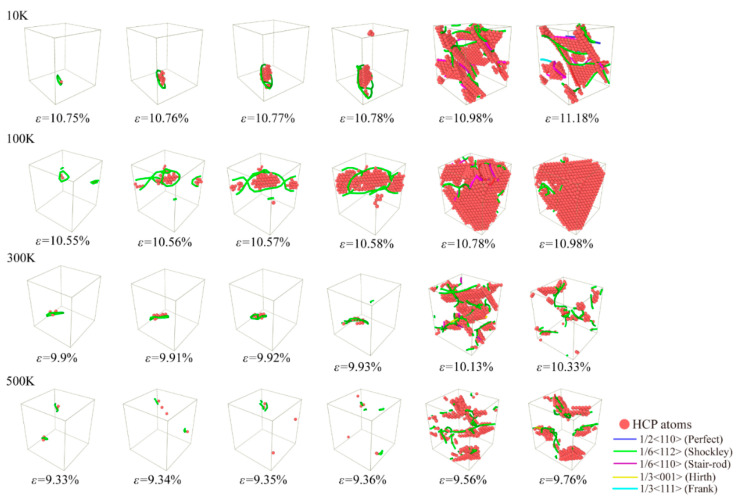
Atomic configuration snapshots of the systems under different tensile strains ε for Al-2%Mg alloy with background temperature span from 10 K to 500 K. The red atoms represent HCP atoms.

**Figure 11 ijms-23-14437-f011:**
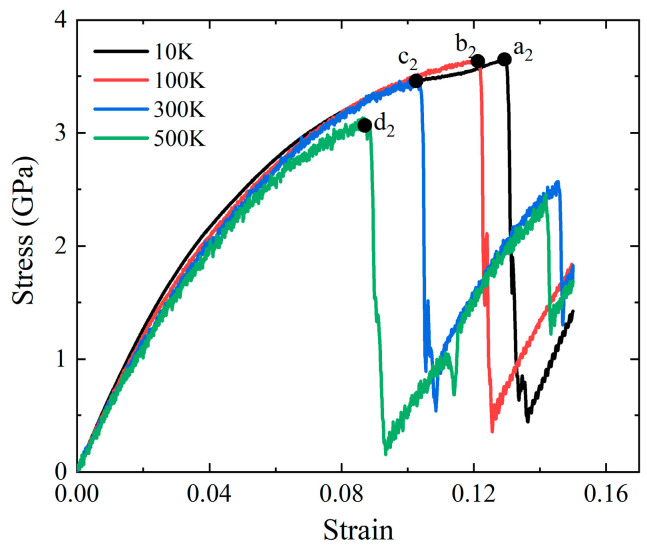
The stress-strain curves for single crystal Al-2%Mg alloys under uniaxial compression with different temperature. The four points, a_2_, b_2_, c_2_ and d_2_, are the maximum stress at 10 K, 100 K, 300 K and 500 K, respectively.

**Figure 12 ijms-23-14437-f012:**
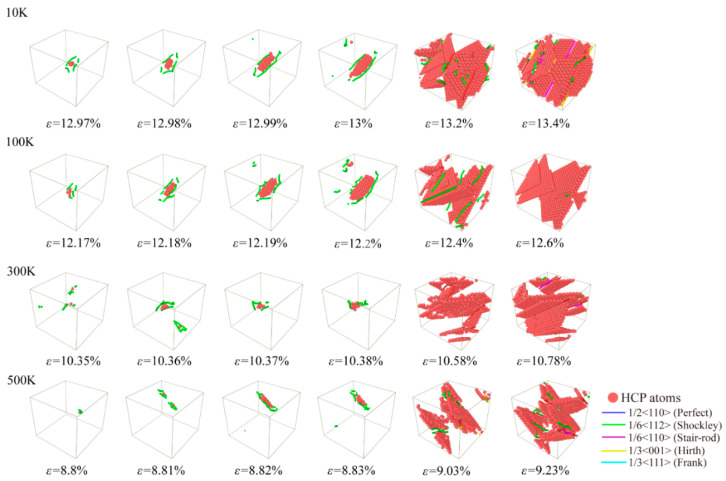
Atomic configuration snapshots of the systems under different compression strains ε for Al-2%Mg alloy with background temperature span from 10 K to 500 K. The red atoms represent HCP atoms.

**Figure 13 ijms-23-14437-f013:**
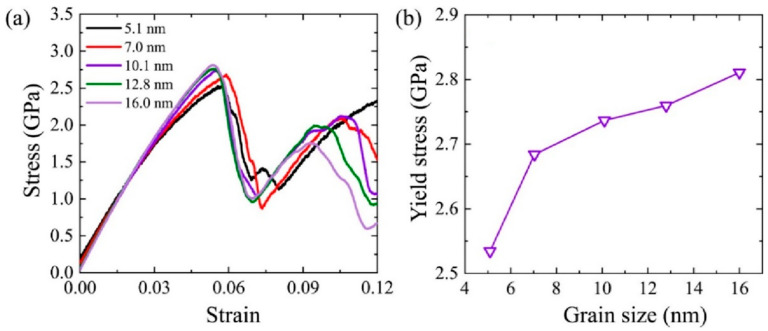
(**a**) Stress-strain curves of polycrystalline Al-Mg alloys with various average grain sizes; (**b**) Relationships between yield stress and grain size at 300 K.

**Figure 14 ijms-23-14437-f014:**
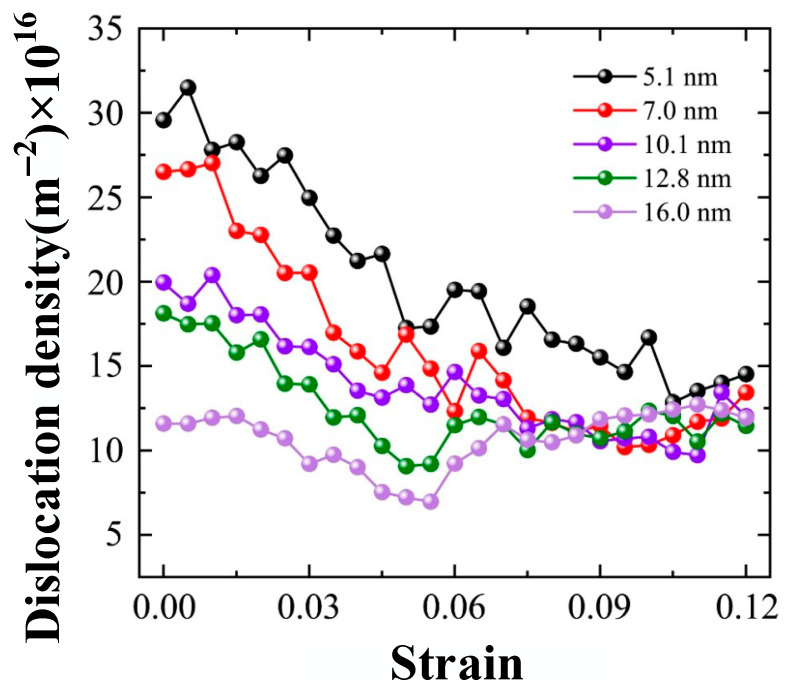
The dislocation density of the system during tensile process of samples with various grain sizes.

**Figure 15 ijms-23-14437-f015:**
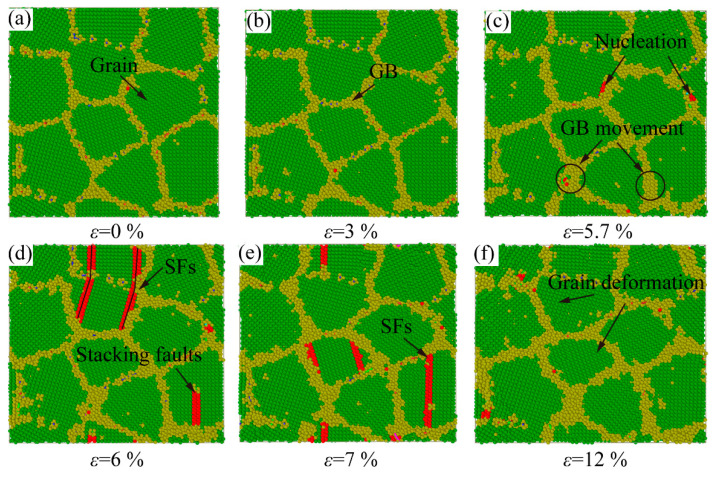
The atomic configurations of polycrystalline Al-2%Mg alloy with grain size D = 5.1 nm at different strains during 300 K tensile simulation. (**a**,**b**) the elastic deformation stage. (**c**−**f**) the movement of GB dominates the plastic deformation stage.

**Figure 16 ijms-23-14437-f016:**
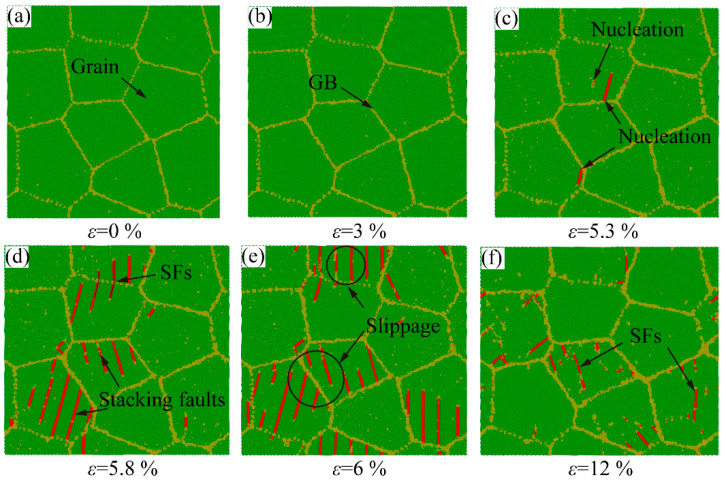
The atomic configurations of polycrystalline Al-2%Mg alloy with grain size D = 16.0 nm at different strains during 300 K tensile simulation. (**a**,**b**) the elastic deformation stage. (**c**−**f**) the dislocation nucleation dominates the plastic deformation.

**Figure 17 ijms-23-14437-f017:**
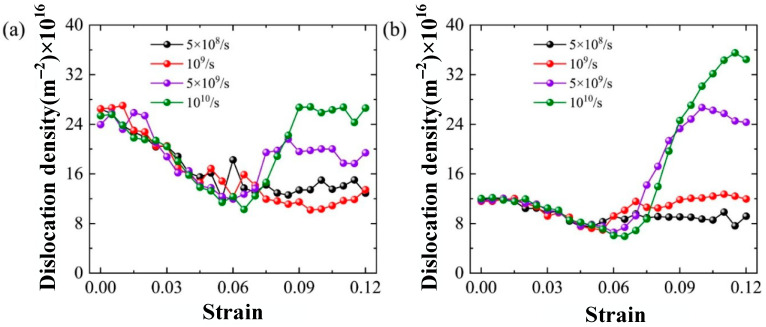
The dislocation density of the samples with various grain sizes. (**a**) grain size d = 7.0 nm, (**b**) grain size d = 16.0 nm.

**Figure 18 ijms-23-14437-f018:**
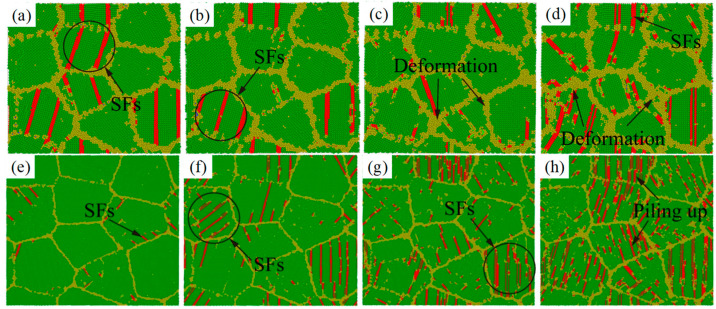
(**a**–**d**) The atomic configurations of the sample with grain size d = 7.0 nm at *ε* = 11% under different strain rates loading. (**a**) 5 × 10^8^ s^−1^, (**b**) 10^9^ s^−1^, (**c**) 5 × 10^9^ s^−1^, (**d**) 10^10^ s^−1^. (**e**–**h**) The atomic configurations of the sample with grain size d = 16.0 nm at *ε* = 11% under various strain rates loading. (**e**) 5 × 10^8^ s^−1^, (**f**) 10^9^ s^−1^, (**g**) 5 × 10^9^ s^−1^, (**h**) 10^10^ s^−1^.

**Figure 19 ijms-23-14437-f019:**
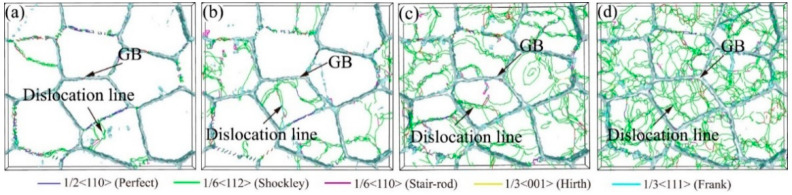
The dislocation lines of the sample with grain size d = 16.0 nm at *ε* = 12% under different strain rates loading. (**a**) 5 × 10^8^ s^−1^, (**b**) 10^9^ s^−1^, (**c**) 5 × 10^9^ s^−1^, (**d**) 10^10^ s^−1^.

**Table 1 ijms-23-14437-t001:** Yield stress and Young’s modulus of defect-free pure Al under uniaxial tension. The work 1 is simulated by Mishin potential, while work 2 is the result of Liu potential.

	Young’s Modulus (GPa)	Yield Stress (GPa)	Potential
Zhu et al. [40]	60	7.8	Mishin [41]
This work1	59	7.8	Mishin [41]
This work2	75	7.2	Liu [29]

**Table 2 ijms-23-14437-t002:** Parameters of models with various grain sizes.

Model	Grain Size (nm)	x × y × z (nm^3^)	Number of Atoms
Model 1	5.1	16.2 × 16.2 × 4.9	74,796
Model 2	7.0	22.3 × 22.3 × 4.9	142,512
Model 3	10.1	32.0 × 32.0 × 4.9	295,944
Model 4	12.8	40.5 × 40.5 × 4.9	475,224
Model 5	16.0	50.6 × 50.6 × 4.9	744,096

## Data Availability

Authors can confirm that all relevant data are included in the article.

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
