# Peer review of "A Multi-Scale Study on Deformation and Failure Process of Metallic Structures in Extreme Environment"

_ijms, 2022, doi:10.3390/ijms232214437_

Round 1

Reviewer 1 Report

This paper describes mainly MD simulations of Mg-Al alloys in simple loading condition. The concept seems based on a multi-scale simulation, but the bridging (linking) method between macroscale and microscale is not clearly explained. The author should adequately describe the way of the multi-scaling for the subject and should explain at least how to configure an adequate boundary condition for microscale analysis from macroscopic simulations. 

The boundary condition shown in Figure.1 (c) and (d)(tensile or compressive loading) seems too simple and not presenting fluid dynamic condition shown in Figure.1 (a) or (b). 

Most of discussion of MD results for alloy system is not new. The authors describes the effect of grain size or strain rate on behavior of crystalline defects of Mg-Al alloys. However, the reviewer finds that the paper only summarizes the difference of results for various conditions, and any relation between the cause and the effect (it should include new findings) is not obviously described. 

The reviewer would like to recommend the authors to reorganize the construction of the paper.

Author Response

Responses to the comments from the reviewers

We greatly appreciate the reviewers and editor for their professional and constructive comments, all comments have important guiding significance for our work. According to the comments, we have carefully revised this manuscript. In the following, we have addressed the reviewer's comments point by point. It should be noted that the reviewer's comments are marked in black italic and the corresponding responses are in blue. All the corrections in the revised manuscript are highlighted in RED.

Reply to the Reviewer #1:

  1. 1. This paper describes mainly MD simulations of Mg-Al alloys in simple loading condition. The concept seems based on a multi-scale simulation, but the bridging (linking) method between macroscale and microscale is not clearly explained. The author should adequately describe the way of the multi-scaling for the subject and should explain at least how to configure an adequate boundary condition for microscale analysis from macroscopic simulations. The boundary condition shown in Figure.1 (c) and (d) (tensile or compressive loading) seems too simple and not presenting fluid dynamic condition shown in Figure.1 (a) or (b).

Reply:

Thanks very much for your comments. In present work, we used a multi-scale method to study the solid-gas interaction system in extreme environments for the first time, and results simulated and predicted the deformation and failure process of truss structures during hypersonic spacecraft reentry. For the linking between the macroscopic and microscopic models, we adopted the simplified model, and obtained the temperature and pressure distribution on its surface through the macroscopic model are loaded into the microscopic system to obtain the microscopic deformation of the material: dislocation, defect initiation and crack propagation. In addition, we compared the temperature and pressure profiles obtained from the macroscopic and microscopic models, as shown in Figure.2. The consistency of macro and micro models is confirmed. We have supplemented the statements of junction between macroscale and microscale models. The confirmation of boundary conditions for microscale are added in the revised manuscript.

In revised manuscript (page 3 line 105 - 114): “The truss structures sustain the structural response to deformation, softening, melting and disintegration with tremendous aerodynamic/thermal loading in Fig. 1(a). In microscale simulation, we employ a similar model to compute the temperature and pressure distribution at the truss structure surface in Fig. 1(b). Fig. 1(c) shows the windward region of the model under a tensile loading. Fig. 1(d) shows the leeward region of the model suffers a compression loading. The initiation and evolution of microstructure defects are dependent on temperatures, loading status and strain rates. With the consideration of the hypersonic reentry aerodynamic environment, we study the effect of temperature, loading pattern and strain rate on truss structure.”

In revised manuscript (page 7 line 245 - 251): “The distribution of temperature and stress provide thermo-mechanical states of metallic structural states under different mechanical and thermal coupling conditions. It shows the windward region under the tensile loading and the leeward region under the compressive loading, respectively. In addition, the boundary conditions of the distribution of temperature and stress are used in the microscale model. With the consideration of the hypersonic reentry aerodynamic environment, we study the effect of temperature, loading pattern and strain rate on truss structure.”

  1. Most of discussion of MD results for alloy system is not new. The authors describe the effect of grain size or strain rate on behavior of crystalline defects of Mg-Al alloys. However, the reviewer finds that the paper only summarizes the difference of results for various conditions, and any relation between the cause and the effect (it should include new findings) is not obviously described.

Reply: Thanks for your comments. The highlight of present work is, it is the first time to use a multi-scale method to study the solid-gas interaction system under extreme environment. The results of temperature and pressure distributions obtained from the macroscale and microscale models are well consistent with each other. It verifies the feasibility and validity of this multi-scale studies. In addition, we also studied the structure of atomic microstructure deformation and failure under different mechanical and thermal coupling effects. The corresponding discussions on the results are supplemented in the revised manuscript.

In revised manuscript (page 13-14 line 415 - 422): “The process of crack propagation is studied under different conducted conditions. The Shockley partial dislocation dominates the deformation in tensile process. The stair-rod dislocation dominates the deformation in compression. Furthermore, the simulations are carried out at various temperature to study the mechanical properties. The increasing temperature decreases the strength under different loading process. Meanwhile, the tensile loading causes less defects comparing with compressive loading. The structure evolution of defects indicates that the alloy strength is more sensitive to compressive loading.”

In revised manuscript (page 20 line 576 - 581): “For polycrystalline Al-Mg alloy model, the yield strength increases with the increasing grain size. The proportion of GB atoms influences the dislocation density. The generation of dislocation decreases the yield strength in small grain size. The plastic deformation dominated by grain boundary sliding in the small grain size, while it is dominated by dislocation slip in the large grain size. The nucleation and propagation of dislocation increase the yield strength at higher strain rate loading process.”

  1. The reviewer would like to recommend the authors to reorganize the construction of the paper.

Reply: Thanks for your comments. We have modified the manuscript structure, statement of results and discussion in the revised version. 

Reviewer 2 Report

I find this paper not suitable for publication in its present form. Characterizing it generally, it presents several independent studies hardly linked with each other. Reading and understanding of paper content is significantly affected by inappropriately low quality of English language. I find interesting and original only the study of the influence of grain structure on the responce of material to tensile and compressive loads (Section 3.3). The authors are suggested to make more detailed investigations in this area and present a paper devoted only to this issue.

More detailed comments can be proposed as follows:

1.The title of the paper is too general. More specific title reflecting the escence of the paper content is required.

2. All the abbrerviations used in the paper must be explained. Please pay attention to explanation of abbreviations both in the Abstract and the main paper text.

3. As I mentioned above, separate investigations carried out in this work are not linked together. The only link is between macroscopic and microscopic results on air flows aroun the console (Fig. 2). However, agreement of these results is not used for further studies. Results obtained for crack in single crystalline Al also stay apart. The same holds for evolution of polycrystalline Al-Mg material.

4. The models are described purely and messy, which does not enable reproduction of the results by other reseachers. In particular, the model for macroscopic simulations is not described at all, only several parameter values without explanation of their origin are provided.

5. Description of the approach to molecular dynamics simulations is not clear. In particular, what potential was used to describe the interaction of Al and Mg atoms? On the one hand, authors provide the parameters for the LJ potential for Al and Mg in lines 126-128. On the other hand, they write about use of EAM in the next lines. It is absolutely unclear how the temperature, pressure as well as elastic characteristics and stress were calculated from the MD simulation results.

6. MD simulation approaches for different subtasks are described in different sections apart from the Section 2 - Models and methods. Either all the model details should be gathered in one section or each subtask should contain model description for it as a subsection. No mixture of these method description modes should be present.

7. I do not understand whether panels (c) and (d) in Fig. 1 present initial input to the model or modeling results. In any case, they are not used further in the paper, so that their relevance is dubious here.

8. How were expressions (2) and (3) obtained? Please provide reference.

9. Section 3.2.1. In lines 211-212 authors write about the model size of about 48x48x48 A. At the same time, they write about grain size of 128 A in lines 222-223. How is it possible?

10. The insets in Fig. 6 do not contain cell orientation, which does not enable to understand the position of crack.

11. Paper contains grammatical errors such as "metal-lic" in the title, "Nose/hoover" instead of "Nose-Hoover" as well as very weird phrases like "microcosmic evolution mechanism". Moreover, the overall quality of English language must be revised. It is not acceptable.

Author Response

Responses to the comments from the reviewers

We greatly appreciate the reviewers and editor for their professional and constructive comments, all comments have important guiding significance for our work. According to the comments, we have carefully revised this manuscript. In the following, we have addressed the reviewer's comments point by point. It should be noted that the reviewer's comments are marked in black italic and the corresponding responses are in blue. All the corrections in the revised manuscript are highlighted in RED.

Reply to the Reviewer #2:

I find this paper not suitable for publication in its present form. Characterizing it generally, it presents several independent studies hardly linked with each other. Reading and understanding of paper content is significantly affected by inappropriately low quality of English language. I find interesting and original only the study of the influence of grain structure on the response of material to tensile and compressive loads (Section 3.3). The authors are suggested to make more detailed investigations in this area and present a paper devoted only to this issue.

Reply: Thanks very much for your positive comment on our work! We have revised the manuscript carefully and accordingly.

  1. The title of the paper is too general. More specific title reflecting the essence of the paper content is required.

Reply: Thanks. We revised the title of manuscript as:

A Prediction of Metallic Structural Failure of Spacecraft: A Multi-Scale Study

In revised manuscript (page 1): “A Prediction of Metallic Structural Failure of Spacecraft: A Multi-Scale Study”

  1. All the abbreviations used in the paper must be explained. Please pay attention to explanation of abbreviations both in the Abstract and the main paper text.

Reply: Thanks, we have checked and corrected the irregular abbreviations in the revised manuscript.

  1. As I mentioned above, separate investigations carried out in this work are not linked together. The only link is between macroscopic and microscopic results on air flows around the console (Fig. 2). However, agreement of these results is not used for further studies. Results obtained for crack in single crystalline Al also stay apart. The same holds for evolution of polycrystalline Al-Mg material.

Reply: Thanks.

In the revised version, we enhanced the connections between the parts in this work. For the linking between the macroscopic and microscopic models, we adopted the simplified model, and obtained the temperature and pressure distribution on its surface through the macroscopic model outflow condition are loaded into the microscopic system to obtain the microscopic deformation of the material: dislocation, defect initiation and crack propagation. In addition, we compared the temperature and pressure profiles obtained from the macroscopic and microscopic models, as shown in Figure.2. The consistency of macro and micro models is confirmed. Moreover, the corresponding discussions are supplemented in the revision.

There have been numerous studies on thermal and mechanical properties of Al-Mg alloy in both macroscale and microscale. However, it is the first time to use a multi-scale method to study the solid-gas interaction system under extreme environment in present study. The results of temperature and pressure distributions obtained from the macroscale and microscale models are well consistent with each other. It verifies the feasibility and validity of this multi-scale studies. In addition, we also studied the structure of atomic microstructure deformation and failure under different mechanical and thermal coupling effects. Based on the mechanical thermal boundary conditions obtained at macroscale model, the failure process of materials under mechanical thermal coupling was systematically studied at microscale. The initiation of defects, the evolution of dislocations, and the structural deformation in single crystal and polycrystalline alloy systems under mechanical and thermal loading were studied. By understanding the mechanism in microscale, it provides a reference for predicting of Al-Mg alloy structure failure. The highlights of present studies are emphasized in revised manuscript. 

In revised manuscript (page 3 line 105 - 114): “We use multi-scale models, from macroscale to microscale, to predict the structural failure of spacecraft truss. The truss structures sustain the structural response to deformation, softening, melting and disintegration with tremendous aerodynamic/thermal loading in Fig. 1(a). In microscale simulation, we employ a similar model to compute the temperature and pressure distribution at the truss structure surface in Fig. 1(b). Fig. 1(c) shows the windward region of the model under a tensile loading. Fig. 1(d) shows the leeward region of the model suffers a compression loading. The initiation and evolution of microstructure defects are dependent on temperatures, loading status and strain rates. With the consideration of the hypersonic reentry aerodynamic environment, we study the effect of temperature, loading pattern and strain rate on truss structure.”

In revised manuscript (page 7 line 245 - 251): “The distribution of temperature and stress provide thermo-mechanical states of metallic structural states under different mechanical and thermal coupling conditions. It shows the windward region under the tensile loading and the leeward region under the compressive loading, respectively. In addition, the boundary conditions of the distribution of temperature and stress are used in the microscale model. With the consideration of the hypersonic reentry aerodynamic environment, we study the effect of temperature, loading pattern and strain rate on truss structure.”

  1. The models are described purely and messy, which does not enable reproduction of the results by other researchers. In particular, the model for macroscopic simulations is not described at all, only several parameter values without explanation of their origin are provided.

Reply: Thanks for your suggestion, we have added the corresponding description of the model in the modified version. For macroscopic simulation, we used the GKUA algorithm to obtain the distribution of temperature and pressure in the outflow field. The required atmospheric environment parameters are the Mach number of incoming-flow, Knudsen number, and specific heat of the gas. Our results of macroscale are well consistent with the previous studies. [Communications in Computational Physics 2016, 20, 773-810.].

In revised manuscript (page 4 line 143 - 149): “For macro-aerodynamic model, the truss structure is set as a beam with the size of 0.015 m × 0.5 m. We simulate the macro mechanical behavior under the hypersonic reentry aerodynamic environment. The exterior of which is the reentry near-continuous transition flow, the inflowing Mach number, Knudsen number and ratio of specific heat are Ma = 8.37, Kn = 0.01 and γ= 1.4, respectively. The exterior flow field of the plate with the mesh of 63×41 in the XOZ coordinate system is computed by the GKUA for solving the Boltzmann model equation with rotational non-equilibrium effect.”

  1. Description of the approach to molecular dynamics simulations is not clear. In particular, what potential was used to describe the interaction of Al and Mg atoms? On the one hand, authors provide the parameters for the LJ potential for Al and Mg in lines 126-128. On the other hand, they write about use of EAM in the next lines. It is absolutely unclear how the temperature, pressure as well as elastic characteristics and stress were calculated from the MD simulation results.

Reply: Thanks for your suggestion, we have added the corresponding description of the molecular dynamics model and simulation parameters in the modified version. In the method section, the parameters of interactions between different types of atoms are added. In the result section, the calculation methods of system temperature, pressure, elastic characteristic and stress are supplemented.

In revised manuscript (page 4-5 line 161 - 172): “The atom interaction can be accurately modeled with the 12-6 Lennard-Jones (LJ) potential for Ar-Ar, Ar-Al and Ar-Mg.

.

(1)

Where ε and σ are the depth of the potential well and the distance at which the LJ interaction is zero, and rij is the distance between two particles. For Ar-Ar interaction, ε = 0.010325 eV and σ = 3.828 Å. For Ar-Mg interaction, ε = 0.0062141 eV and σ = 3.0687 Å. For Ar-Al interaction, an LJ potential with parameters ε = 0.01325 eV and σ = 3.7271 Å is used as a reference potential. The LJ potential cutoff radius is set as 10 Å.

The EAM potential is used to described Al-Al, Al-Mg and Mg-Mg interactions of Al-Mg alloys.

 .

(2)

Where rij is the separation distance between atoms i to j, Uα is the energy required to embed an atom in the electron cloud, and ρ(rij) is the electron transfer function between the atoms. The pairwise part Φ(rij) is the interaction of the atom i to j.”

In revised manuscript (page 5 line 189 - 198): “The number density, temperature and pressure distribution solved by MD simulation in Fig. 2(c-e). Microscopic observables, including temperature T and pressure P are calculated as the spatial and time average in the volume of a prescribed grid cell by the following formula [Mechanics of Materials 2020, 150, 103588]:

.

(5)

.

(6)

Where k is the Boltzmann constant, N is the number of atoms, T is the temperature, V is the volume of grid cell, ma is the mass, va is the velocity, ra is the position vector and Fa is the force vector of the atom a. The volume of the grid cell with a size defined as Δx × Δy × Δz. When drawing the outflow distribution in the x-z plane, the size of Δx × Δz is set as 1 Å × 1 Å for density and temperature, and 2 Å × 2 Å for pressure. The length of Δy is set as the length of the box in the y-direction.”

In revised manuscript (page 8 line 282 - 284): “The Young’s modulus is obtained by linear fitting of strain ε = 0.6% - 1.4% [Key Engineering Materials 2019, 804, 1-6] in the elastic stage of the stress-strain curve. During the tensile simulation, the point of maximum stress is considered as the yield stress.”

  1. MD simulation approaches for different subtasks are described in different sections apart from the Section 2 - Models and methods. Either all the model details should be gathered in one section or each subtask should contain model description for it as a subsection. No mixture of these method description modes should be present.

Reply: Thanks for your suggestion, we modified the related descriptions that may cause ambiguity or misunderstanding in the revised version. The corresponding descriptions of the molecular simulation models are supplemented in the related subtasks.

In revised manuscript (page 3 line 120 - 128): “The macroscale and microscale simulation details for the distribution of temperature and pressure are described in section 3.1.1. In section 3.2.1, a single crystalline Al-2%Mg model is applied to study the microstructure evolution of defects under different loading pattern and various temperature. Moreover, a polycrystalline Al-2%Mg alloy model is carried out a simulation to study the mechanical properties. In section 3.2.1, A pure Al model and a model with a central crack are used to study the defects generation and the propagation of cracks under different loading pattern. In section 3.3.1, five polycrystalline Al-2%Mg alloy models with various grain size are carried out a tensile simulation to study the effect of strain rate.”

  1. I do not understand whether panels (c) and (d) in Fig. 1 present initial input to the model or modeling results. In any case, they are not used further in the paper, so that their relevance is dubious here.

Reply: Thanks very much for your comments. In present work, we used a multi-scale method to study the solid-gas interaction system in extreme environments for the first time, and results simulated and predicted the deformation and failure process of truss structures during hypersonic spacecraft reentry. For the linking between the macroscopic and microscopic models, we adopted the simplified model, and obtained the temperature and pressure distribution on its surface through the macroscopic model outflow condition are loaded into the microscopic system to obtain the microscopic deformation of the material: dislocation, defect initiation and crack propagation. Here, Figure. 1(a) and (b) show the aerodynamic conditions of the spacecraft during hypersonic flight, respectively. The temperature and pressure distributions obtained from the macroscopic simulations are inset into the molecular dynamics model as boundary conditions for the microscopic simulations. By means of the all-atom model of molecular dynamics simulation, the microscopic mechanism of material failure under different mechanical and thermal coupling conditions is studied. In the revised version, we added a description of the combination of macroscale and microscale models. As the first multi-scale study of gas-solid interface, the model is still rough. We will refine this model in the future study.

In revised manuscript (page 3 line 105 - 114): “We use multi-scale models, from macroscale to microscale, to predict the structural failure of spacecraft truss. The truss structures sustain the structural response to deformation, softening, melting and disintegration with tremendous aerodynamic/thermal loading in Fig. 1(a). In microscale simulation, we employ a similar model to compute the temperature and pressure distribution at the truss structure surface in Fig. 1(b). Fig. 1(c) shows the windward region of the model under a tensile loading. Fig. 1(d) shows the leeward region of the model suffers a compression loading. The initiation and evolution of microstructure defects are dependent on temperatures, loading status and strain rates. With the consideration of the hypersonic reentry aerodynamic environment, we study the effect of temperature, loading pattern and strain rate on truss structure.”

  1. How were expressions (2) and (3) obtained? Please provide reference.

Reply: Thanks. We have supplemented the corresponding references in the revised manuscript.

In revised manuscript (page 5 line 183 - 188): “Therefore, we can obtain the actual temperature Ta and Pa values conducted to the plate boundary through the following formula [Communications in Computational Physics 2016, 20, 773-810]:

.

(3)

.

(4)

Where T, ρ and V represent the reference temperature, gas density and velocity of the external flow field, respectively. T and P represent the temperature and pressure values of the external flow field calculated by GKUA, respectively.”

  1. Section 3.2.1. In lines 211-212 authors write about the model size of about 48x48x48 A. At the same time, they write about grain size of 128 A in lines 222-223. How is it possible?

Reply: Thanks. We have modified the ambiguous and misleading statements in the revised manuscript. For the single crystal model, its dimensions are 48×48×48 A3, while they are 405 × 405 × 49 A3 for the polycrystalline model.

In revised manuscript (page 7 line 256 - 257): “For the single crystal model, its dimensions are 48×48×48 A3, while they are 405 × 405 × 49 A3 for the polycrystalline model.”

  1. The insets in Fig. 6 do not contain cell orientation, which does not enable to understand the position of crack.

Reply: Thanks. We have added the coordinate directions in the revised manuscript.

In revised manuscript (page 9):

Figure 6. Stress-strain curve and dislocation analysis of pure Al model under different loading process (a) Uniaxial tension and (b) Uniaxial compression. The insets figures present the atomic structure by common neighbor analysis for the dislocation emitted under tension and compression loading. The green and red atoms represent FCC and HPC atoms, respectively.

  1. Paper contains grammatical errors such as "metal-lic" in the title, "Nose/hoover" instead of "Nose-Hoover" as well as very weird phrases like "microcosmic evolution mechanism". Moreover, the overall quality of English language must be revised. It is not acceptable.

Reply: Thanks. We have modified the grammatical/statement/spelling errors in the revised manuscript and the quality of article language are improved as possible.

Reviewer 3 Report

In this paper, the authors using multi-scale methods explored defect initiation, growth and crack propagation of metallic truss structure under high engine temperature and pressure conditions. The distributions of temperature and strain field in the aerodynamic environment obtained by molecular dynamics simulations are in good agreement with those obtained from the macroscopic Boltzmann method. Overall, the paper is well organized, and the analysis is also well supported from the calculated data and the revised manuscript has been improved. I recommend acceptance of the paper after a few minor revisions:

 (1)  The authors are suggested to provide more detailed discussion on how to bridge the link between macroscale and microscale simulation.

(2)  Line 193, the meaning of each variables/characters in equation 6 should be clearly defined.

Author Response

  Reply to the Reviewer #3: In this paper, the authors using multi-scale methods explored defect initiation, growth and crack propagation of metallic truss structure under high engine temperature and pressure conditions. The distributions of temperature and strain field in the aerodynamic environment obtained by molecular dynamics simulations are in good agreement with those obtained from the macroscopic Boltzmann method. Overall, the paper is well organized, and the analysis is also well supported from the calculated data and the revised manuscript has been improved. I recommend acceptance of the paper after a few minor revisions: 1. The authors are suggested to provide more detailed discussion on how to bridge the link between macroscale and microscale simulation. Reply: Thanks for your comments. In the revised version, we enhanced the connections between the parts in this work. For the linking between the macroscopic and microscopic models, we adopted the simplified model, and obtained the temperature and pressure distribution on its surface through the macroscopic model outflow condition are loaded into the microscopic system to obtain the microscopic deformation of the material: dislocation, defect initiation and crack propagation. In addition, we compared the temperature and pressure profiles obtained from the macroscopic and microscopic models, as shown in Figure.2. The consistency of macro and micro models is confirmed. Moreover, the corresponding discussions are supplemented in the revision. There have been numerous studies on thermal and mechanical properties of Al-Mg alloy in both macroscale and microscale. However, it is the first time to use a multi-scale method to study the solid-gas interaction system under extreme environment in present study. The results of temperature and pressure distributions obtained from the macroscale and microscale models are well consistent with each other. It verifies the feasibility and validity of this methods. In addition, we also studied the structure of atomic microstructure deformation and failure under different mechanical and thermal coupling effects. Based on the mechanical thermal boundary conditions obtained at macroscale model, the failure process of materials under mechanical thermal coupling was systematically studied at microscale. The initiation of defects, the evolution of dislocations, and the structural deformation in single crystal and polycrystalline alloy systems under mechanical and thermal loading were studied. By understanding the mechanism in microscale, it provides a reference for predicting of Al-Mg alloy structure failure. The highlights of present studies are emphasized in revised manuscript. In revised manuscript (page 7 line 262 - 268): “The distribution of temperature and stress provide thermo-mechanical states of metallic structural states under different mechanical and thermal coupling conditions. It shows the windward region under the tensile loading and the leeward region under the compressive loading, respectively. In addition, the boundary conditions of the distribution of temperature and stress are used in the microscale model. With the consideration of the hypersonic reentry aerodynamic environment, we study the effect of temperature, loading pattern and strain rate on truss structure.” In revised manuscript (page 14 line 449 - 453): “During the hypersonic reentry aerodynamic environment, the windward and leeward regions suffer different temperature and loading condition. The effect of temperature and loading pattern are systematically analyzed by MD simulation. Those results can provide an atomistic-scale mechanism on the structure failure by defects evolution.” In revised manuscript (page 20 line 604 - 608): “In addition, the strain rate corresponds to the boundary conditions of the distribution of stress on the spacecraft surface. Large pressure on surface means a high strain rate of samples. Those results can understand the mechanism of structure failure under hyperthermal pressure condition.” 2. Line 193, the meaning of each variable/characters in equation 6 should be clearly defined. Reply: Thanks for your comments. We have added the corresponding description of each variable/character in equation (6). In revised manuscript (page 5 line 193 - 198): “Microscopic observables, including temperature T and pressure P are calculated as the spatial and time average in the volume of a prescribed grid cell by the following formula: . (5) . (6) Where T is the temperature, k is the Boltzmann constant, N is the number of atoms, V is the volume of grid cell, ma is the mass, va is the velocity vector, ra is the position vector, and Fa is the force vector of the atom a. Pij is the pressure tensor (where i and j are x, y, or z).”

Round 2

Reviewer 1 Report

Some descriptions have been added and the paper is improved.

But, after reading through your manuscript, the reviewer would like to clarify the following points.

(1)   As to the equations (3) and (4) (page 21), physical dimensions of temperature and pressure seems inconsistent. For example, the unit of right hand side of equation (3) should be Kelvin squared, which does not match to left hand side. Please clarify the physical dimension of each components in the text.

(2)   In line 227, The authors state that “macroscale and microscale models are well consistent with each other”. Also, in line 216, “the flow field results are in good agreement”. However, the agreement is not clear to the reviewer and maybe to the readers. Please point out  the agreement more clearly, with regards to Figure 2 (a)(b)(macro) and (c)(d)(e)(micro).

(3)   In displaying coordinates, upper case “X” “Y” “Z” are used as well as “x” “y” “z”. They should be unified to one of them.

(4)   Figure 3(page 7), “solide” should read “solid”.

(5)   In section 3.2, the authors conducted the loading simulations. They used 3-dimensional periodic boundary conditions. After that, they also evaluate Young’s modulus, which should be uniaxial conditions. The authors should clearly describe how uniaxial condition is realized with periodic conditions.

(6)   Figure 10 (maybe for tensile loading) and Figure 12 (maybe for compression one) are not discerned because their captions are the same. The authors should add loading condition to each figure caption.

(7)   Resolution of Figure 10 and 12 are not enough to look atomic movement of defects. It would be grateful if the author improve the figure (size, the number of figures, resolution, …).

(8)   Line 580(page 19), “The plastic deformation dominated by …” should read “The plastic deformation *is* dominated by…”. Please check especially the correctness of sentences added in revision.

Author Response

Responses to the comments from the reviewers

We greatly appreciate the reviewers and editor for their professional and constructive comments, all comments have important guiding significance for our work. According to the comments, we have carefully revised this manuscript. In the following, we have addressed the reviewer's comments point by point. It should be noted that the reviewer's comments are marked in black italic and the corresponding responses are in blue. All the corrections in the revised manuscript are highlighted in RED.

Reply to the Reviewer #1:

  1. As to the equations (3) and (4) (page 21), physical dimensions of temperature and pressure seems inconsistent. For example, the unit of right hand side of equation (3) should be Kelvin squared, which does not match to left hand side. Please clarify the physical dimension of each component in the text.

Reply: Thanks for your comments. We have modified the ambiguous and misleading statements in the revised manuscript. The T and P in the right side of equation (3) and (4) are dimensionless components.

In revised manuscript (page 5 line 183 - 189): “Therefore, we can obtain the actual temperature Ta and Pa values conducted to the plate boundary through the following formula [Communications in Computational Physics 2016, 20, 773-810]:

.

(3)

.

(4)

Where T, ρ and V represent the reference temperature, gas density and velocity of the external flow field, respectively. T and P represent the temperature and pressure parameters of the external flow field calculated by GKUA, respectively. The T and P in the right side of equation (3) and (4) are dimensionless components.”

  1. In line 227, The authors state that “macroscale and microscale models are well consistent with each other”. Also, in line 216, “the flow field results are in good agreement”. However, the agreement is not clear to the reviewer and maybe to the readers. Please point out the agreement more clearly, with regards to Figure 2 (a)(b)(macro) and (c)(d)(e)(micro).

Reply: Thanks very much for your comments. In present work, we used a multi-scale method to study the solid-gas interaction system in extreme environments for the first time. Between the macroscopic and microscopic models, we adopted the simplified model, and obtained the temperature and pressure distribution on its surface through the macroscopic model and the microscopic system. We obtained the similar phenomenons. The distribution and variation of temperature and pressure are in good agreement in macroscale and microscale results. As shown in Fig.2, the incoming hypersonic flow in the near-continuum transition flow crosses the strongly disturbed detached shock wave layer in front of the stagnation points region, reaching the high temperature and high pressure around the plate. The airflow expands rapidly into the leeward region. The temperature and pressure decrease to the lowest at the rear stagnation point. The highest temperature appears at the left top inflection point of the plate. It has the maximum curvature variation and is subjected to the most severe aerothermodynamic heating. The maximum pressure appears at the bottom of the plate due to the increasing atomic density leads to the compression of gas. The consistency of macro and micro models is confirmed.

In revised manuscript (page 6 line 229 - 241): “Between the macroscopic and microscopic models, we adopted the simplified model, and obtained the temperature and pressure distribution on its surface through the macroscopic model and the microscopic system. We obtained the similar phenomenons. The distribution and variation of temperature and pressure are in good agreement in macroscale and microscale results. As shown in Fig.2, the incoming hypersonic flow crosses the disturbed detached shock wave layer in front of the stagnation points region, reaching the high temperature and high pressure around the plate. The airflow expands into the leeward region rapidly. The temperature and pressure decrease to the lowest at the rear stagnation point. The highest temperature appears at the left top inflection point of the plate. It has the maximum curvature variation and is subjected to the most severe aerothermodynamic heating. The maximum pressure appears at the bottom of the plate due to the increasing atomic density leads to the compression of gas. The consistency of macro and micro models is confirmed.”

  1. In displaying coordinates, upper case “X” “Y” “Z” are used as well as “x” “y” “z”. They should be unified to one of them.

Reply: Thanks for your comments. We have unified the displaying coordinates to upper case “X” “Y” and “Z” in the revised manuscript.

In revised manuscript (page 6 line 216 - 217): “The windward boundary temperature of the plate rises gradually along the positive Z-direction in Fig. 2(d).”

In revised manuscript (page 6 line 216 - 217): “The temperature near the windward stagnation point reaches the maximum value in X-direction.”

In revised manuscript (page 7 line 246 - 248): “When plotting the curves of microscopic parameters along the Z-direction, we set the length of Δz as 10 Å and the size of Δx × Δy is 20 Å × 40.5 Å.”

In revised manuscript (page 7 line 271 - 272): “The single crystal orientation of X, Y, and Z directions are [100], [010], and [001], respectively.”

In revised manuscript (page 8 line 281 - 285): “

Figure 4. Molecular dynamics simulation models for Al-Mg alloys with 2% Mg content. (a) The single crystal Al-Mg alloy model. The red atoms denote aluminum atom, and the blue atoms denote magnesium atom. (b) The polycrystalline Al-Mg alloy model with grain size d = 12.8 nm. The green atoms represent grain, and the yellow atoms represent GB.”

  1. Figure 3(page 7), “solide” should read “solid”.

Reply: Thanks for your comments. We have modified the errors in the revised manuscript.

In revised manuscript (page 7 line 242 - 244): “

Figure 3. Pressure and temperature distribution along the plate surface solved by MD simulation. The blue square denote pressure, and the red circle denote temperature.”

  1. In section 3.2, the authors conducted the loading simulations. They used 3-dimensional periodic boundary conditions. After that, they also evaluate Young’s modulus, which should be uniaxial conditions. The authors should clearly describe how uniaxial condition is realized with periodic conditions.

Reply: Thanks for your suggestion, we have added the corresponding description of uniaxial condition in modified version.

In revised manuscript (page 8 line 293 - 299): “Uniform uniaxial tension with constant strain rate was modeled by the scaling of coordinates of atoms with the use of ‘deform’ command of LAMMPS. The uniaxial tensile simulation is to squeeze the material throughout its length which leads to elongation along the Z-direction. This can also be visualized as a change in the simulation box size to deform the material. The periodic boundary conditions are applied in all directions. The interaction of dislocation with other dislocations, GB, and precipitates are usually studied using this method.”

  1. Figure 10 (maybe for tensile loading) and Figure 12 (maybe for compression one) are not discerned because their captions are the same. The authors should add loading condition to each figure caption.

Reply: Thanks for your comments. We have added the corresponding loading condition in the revised manuscript. Figure 10 (tensile loading) and Figure 12 (compressive loading) are revised figure caption.

In revised manuscript (page 12 line 409 - 411): “Figure 10. Atomic configuration snapshots of the systems under different tensile strains ε for Al-2%Mg alloy with background temperature span from 10 K to 500 K.”

In revised manuscript (page 13 line 435 - 437): “Figure 12. Atomic configuration snapshots of the systems under different compression strains ε for Al-2%Mg alloy with background temperature span from 10 K to 500 K.”

  1. Resolution of Figure 10 and 12 are not enough to look atomic movement of defects. It would be grateful if the author improve the figure (size, the number of figures, resolution, …).

Reply: Thanks for your comments. We have improved figures quality in the revised version.

In revised manuscript (page 12 line 408 - 411): “

Figure 10. Atomic configuration snapshots of the systems under different tensile strains ε for Al-2%Mg alloy with background temperature span from 10 K to 500 K. The red atoms represent HCP atoms.”

In revised manuscript (page 13 line 434 - 437): “

Figure 12. Atomic configuration snapshots of the systems under different compression strains ε for Al-2%Mg alloy with background temperature span from 10 K to 500 K. The red atoms represent HCP atoms.”

  1. Line 580(page 19), “The plastic deformation dominated by …” should read “The plastic deformation *is* dominated by…”. Please check especially the correctness of sentences added in revision.

Reply: Thanks for your comments. We have modified the grammatical errors in the revised manuscript and the quality of article language are improved as possible.

In revised manuscript (page 19 line 601 - 603): “The plastic deformation is dominated by grain boundary sliding in the small grain size, while it is dominated by dislocation slip in the large grain size.”

Reviewer 2 Report

I am balancing my decision on this paper between  major revision and complete rejection. Although the authors have made a great effort to answer my comment and a lot of technical details have become clear now, the paper is still of very low quality and does not correspond to authors' claims. In particular, authors still present several independent researches not properly linked each to other. When one uses multiscale approach, the results and the values obtained at one scale are further used at different scale. No such approach is realized here. It is demonstrated that both macroscopic and MD methods provide similar distributions of pressure and temperature around the bar. This result is used no more. Next, generation and evolution of dislocations and crack in Al material under the action of tensile and compressive strain at different temperatures is modeled. This is quite independent problem from the previous one as no results of it are used. To my opinion, the results obtained here are quite obvious and not interesting therefore. Finally, I find interesting the results on the evolution of grains and extended defects in polycrystalline material. But again, this is quite independent problem from all the others that may be considered individually.

Summarizing this part, I believe that provided authors make further modifications of the paper to make it publishable, they should avoid use of terminology "multiscale" as their research does not correspond to what is usually meant by it.

Other comments include:

1. The title should be informative and relevant to the paper content. Although their study may be useful for spacecraft engineering, the authors do not consider failure of spacecraft but only of Al material.

2. Quality of English language is extremely low and must be thoroughly improved.

Author Response

Reply to the Reviewer #2: I am balancing my decision on this paper between major revision and complete rejection. Although the authors have made a great effort to answer my comment and a lot of technical details have become clear now, the paper is still of very low quality and does not correspond to authors' claims. In particular, authors still present several independent researches not properly linked each to other. When one uses multiscale approach, the results and the values obtained at one scale are further used at different scale. No such approach is realized here. It is demonstrated that both macroscopic and MD methods provide similar distributions of pressure and temperature around the bar. This result is used no more. Reply: Thanks very much for your comment! Till now, a systematical study on the solid-gas interaction in extreme environments using multi-scale method is still lacking. In present work, we used multi-scale method for the exploratory research on the solid-gas interaction system. Here, the temperature and pressure distribution of the truss structure was obtained from macroscopic model (a GKUA model). In order to obtain the microstructure deformation at corresponding macroscopic condition, we studied the initiation of defects, the generation and evolution of dislocations at specific boundary conditions using MD simulations. We will broaden our research through further coupling the parameters with the consideration of larger MD model in our further studies. In revised manuscript (page 7 line 262 - 268): “The distribution of temperature and stress provide thermomechanical states of metallic structural states under different mechanical and thermal coupling conditions. It shows the windward region under the tensile loading and the leeward region under the compressive loading, respectively. In addition, the boundary conditions of the distribution of temperature and stress are used in the microscale model. With the consideration of the hypersonic reentry aerodynamic environment, we study the effect of temperature, loading pattern and strain rate on truss structure.” In revised manuscript (page 14 line 449 - 453): “During the hypersonic reentry aerodynamic environment, the windward and leeward regions suffer different temperature and loading condition. The effect of temperature and loading pattern are systematically analyzed by MD simulation. Those results can provide an atomistic-scale mechanism on the structure failure by defects evolution.” In revised manuscript (page 20 line 604 - 608): “In addition, the strain rate corresponds to the boundary conditions of the distribution of stress on the spacecraft surface. Large pressure on surface means a high strain rate of samples. Those results can understand the mechanism of structure failure under hyperthermal pressure condition.” Next, generation and evolution of dislocations and crack in Al material under the action of tensile and compressive strain at different temperatures is modeled. This is quite independent problem from the previous one as no results of it are used. To my opinion, the results obtained here are quite obvious and not interesting therefore. Reply: We computed the deformation and pressure distribution around the truss structure through macroscopic model (the GKUA model). In order to obtain the microstructure deformation such as the initiation of defects, the generation and evolution of dislocations, we conducted MD simulations at the specific boundary conditions. A beam model with size of 12.4 nm was employed. The tension and compression on the bending surface of the truss structure can be approximated as uniaxial tensile and compressive deformation. In revised manuscript (page 3 line 108 - 114): “In microscale simulation, we employ a beam model to compute the temperature and pressure distribution at the truss structure surface in Fig. 1(b). Fig. 1(c) shows the windward region of the model under a tensile loading. Fig. 1(d) shows the leeward region of the model suffers a compression loading. The initiation and evolution of microstructure defects are dependent on temperatures, loading status and strain rates. With the consideration of the hypersonic reentry aerodynamic environment, we study the effect of temperature, loading pattern and strain rate on truss structure.” In revised manuscript (page 14 line 449 - 453): “During the hypersonic reentry aerodynamic environment, the windward and leeward regions suffer different temperature and loading condition. The effect of temperature and loading pattern are systematically analyzed by MD simulation. Those results can provide an atomistic-scale mechanism on the structure failure by defects evolution.” Finally, I find interesting the results on the evolution of grains and extended defects in polycrystalline material. But again, this is quite independent problem from all the others that may be considered individually. Reply: Thanks for your comments. In the manuscript, we considered the properties of the defect evolution and dislocation propagation in polycrystalline GB at various stress and temperature boundary condition. In addition, we also investigated the properties of microstructure evolution at different stress and temperature boundary condition according to the macroscopic model. For the size effect on nanoscale model, the coupling parameters at different scale is still uncertain. We will explore the larger model to test the parameter size-dependent effects. In revised manuscript (page 20 line 600 - 609): “For polycrystalline Al-Mg alloy model, the yield strength increases with the increasing grain size. The proportion of GB atoms influences the dislocation density. The generation of dislocation decreases the yield strength in small grain size. The plastic deformation is dominated by grain boundary sliding in the small grain size, while it is dominated by dislocation slip in the large grain size. The nucleation and propagation of dislocation in-crease the yield strength at higher strain rate loading process. In addition, the strain rate corresponds to the boundary conditions of the distribution of stress on the spacecraft sur-face. Large pressure on surface means a high strain rate of samples. Those results can understand the mechanism of structure failure under hyperthermal pressure condition.” Summarizing this part, I believe that provided authors make further modifications of the paper to make it publishable, they should avoid use of terminology "multiscale" as their research does not correspond to what is usually meant by it. Reply: Thanks for your comments. We have added corresponding discussions on “multiscale” in the modified version. In revised manuscript (page 7 line 262 - 268): “The distribution of temperature and stress provide thermo-mechanical states of metallic structural states under different mechanical and thermal coupling conditions. It shows the windward region under the tensile loading and the leeward region under the compressive loading, respectively. In addition, the boundary conditions of the distribution of temperature and stress are used in the microscale model. With the consideration of the hypersonic reentry aerodynamic environment, we study the effect of temperature, loading pattern and strain rate on truss structure.” In revised manuscript (page 14 line 449 - 453): “During the hypersonic reentry aerodynamic environment, the windward and leeward regions suffer different temperature and loading condition. The effect of temperature and loading pattern are systematically analyzed by MD simulation. Those results can provide an atomistic-scale mechanism on the structure failure by defects evolution.” In revised manuscript (page 20 line 604 - 608): “In addition, the strain rate corresponds to the boundary conditions of the distribution of stress on the spacecraft surface. Large pressure on surface means a high strain rate of samples. Those results can understand the mechanism of structure failure under hyperthermal pressure condition.” Other comments include: 1. The title should be informative and relevant to the paper content. Although their study may be useful for spacecraft engineering, the authors do not consider failure of spacecraft but only of Al material. Reply: Thanks for your comments. We revised the title of manuscript to be more informative and relevant to the paper content. In revised manuscript (page 1): “A Macro-Microscale Study on Deformation and Failure Process of Metallic Structures in Aerodynamic Hyperthermal Environment” 2. Quality of English language is extremely low and must be thoroughly improved. Reply: Thanks. We have modified the grammatical/statement/spelling errors in the revised manuscript and the quality of article language are improved as possible. In revised manuscript (page 1 line 39 - 41): “As a preliminary research for above issues, there is practical significance to develop the numerical forecast and analysis for predicting disintegration. It reduces the risk posed to population and the property on the ground.” In revised manuscript (page 5 line 178 - 183): “Because of the reentry near-continuum transition flow region, the thick detached shock wave layer is formed around the thin plate. The strong aerodynamic heating and force are imposed as the interface condition of the present thermodynamic on the plate surface. The incoming hypersonic flow crosses the strongly disturbed detached shock wave layer in front of the stagnation points region, reaching the highest temperature and stress.” In revised manuscript (page 7 line 256 - 259): “On the one hand, the compression of gas causes the increase of atom density in the left bottom surface. On the other hand, the airflow expands rapidly into the leeward region after through the corner of the plate resulting in a decrease in pressure.” In revised manuscript (page 9 line 337 - 341): “From the dislocation analysis in Fig. 6(b), dislocations emitted mean that the system has entered the plastic deformation stage. The Shockley dislocations and the Stair-rod dislocations dominate the plastic deformation at compression loading. The entanglement of SFs generates the stair-rod dislocations between two slip planes, as shown the atomic configuration in black circle.” In revised manuscript (page 10 line 347 - 357): “The statistics analysis of HCP atoms and dislocation lines at two loading patterns are illustrated in Fig. 7. The curves of HCP atom percentage and dislocation density with the strain at tensile loading are monitored in Fig. 7 (a,b), while the curves at compression loading is presented in Fig. 7(c,d). A similar result can be found between tensile and compressive loading. The ratio of HCP atom increases sharply in the plastic stage, and de-clines with the increase of strain after reaching the peak value. In the meanwhile, the peak points of the HCP atom ratio and dislocation density are appeared at a same strain. Obvious peak values can be observed from Fig. 7 at different loading. A peak value of the ratio HCP atom at compression condition is found around 9%. This is almost 9 times greater than the value at tensile process. However, the compressive process obtains a smaller peak value of dislocation density.” In revised manuscript (page 11 line 361 - 375): “Fig. 8 represents the curve of the crack length with strain in the pure Al model and the shape of the crack propagation under tensile loading. The growth process of crack is roughly subdivided into three different stages from Fig. 8(a). An elastic deformation is shown in stage I. A stable crack length is observed in this stage. There is no dislocation around crack in Fig. 8(b), indicated a stable lattice structure. In stage Ⅱ, the material enters the plastic deformation stage. The length of crack increases from 2.74 to 7.23 nm sharply when the strain increases. The maximum central symmetry parameter (CSP) value is observed near the crack tip, meaning the generation of initial defects. The inset shows the atomic structure around crack by Common Neighbor Analysis (CNA). From Fig. 8(c), defects are consisted of SFs and disorder atoms, resulting in the plastic deformation (as shown in the inset figure, where red atoms represent the SFs, green atoms represent the FCC structure, and white atoms represent the disordered structure). In stage III, the crack grows slowly. From Fig. 8(d), disordered structures around crack tip inhibit further crack growth thus increasing the critical stress for crack propagation. Defects are distributed from crack tip to the whole system.” In revised manuscript (page 11 line 381 - 393): “Tensile simulations of the mechanical properties and failure mechanism of Al-2%Mg single crystal alloy are carried out at temperatures ranging from 10K to 500K. Stress-strain curves of Al-2%Mg alloys at different temperatures are plotted in Fig. 9. The four curves show a linear relationship before reaching the yield point. Four points, a1, b1, c1 and d1, are the maximum stress at 10 K, 100K, 300 K and 500 K, respectively. After the point of maximum stress, the stress drops simultaneously as the strain increases, showing the generation and growth of defects, which was considered as failure in the works. As the strain in-creases, the samples enter the plastic deformation stage. From Fig. 9, the effect of temperature on yield stress is significant. At 10K-500K, the yield stress decrease sharply from 9.12 GPa to 5.43 GPa. As the temperature increases, the yield stress decrease due to more phonons absorbed by lattice, the increased atomic mobility and the rapid diffusion of free volume. Hence, the atoms can be displaced more easily at higher temperature when the external loading is applied.” In revised manuscript (page 12 line 397 - 407): “Uniaxial tensile loading is carried out to investigate the evolution mechanism of microstructure for Al-2%Mg single crystal alloy under different temperatures. In Fig. 10, the atomic structure snapshots present the defects evolution at every 0.01% strain from the yield point under four temperatures condition (10 K, 100 K, 300 K and 500 K). Corresponding to the stress-strain curves in Fig. 9, the snapshots appear initial dislocation at the strain of the yield points. As the temperature increases, the strain of initial dislocation emission decreases obviously. In addition, we analyze the initial dislocation types at different temperature, Shockley dislocations are observed in all samples. The dislocations begin to grow and slide when the strain increases. At high temperature, the strength de-creases depending on the less defective structures and short dislocation lines. Meanwhile, higher proportion of HCP atoms strengthens the structure stability at low temperature.” In revised manuscript (page 13 line 413 - 422): “Uniaxial compression simulations of the mechanical properties and failure mechanism of Al-2%Mg single crystal alloy are carried out at temperatures ranging from 10K to 500K. Stress-strain curves of Al-2%Mg alloys at different temperatures are plotted in Fig. 11. Four curves show a nonlinear relationship before reaching the yield point. Four points, a1, b1, c1 and d1, are the maximum stress at 10 K, 100K, 300 K and 500 K, respectively. As the increase of strain, the samples enter the plastic deformation stage and the stress drops sharply. The increase of temperature decreases the yield strength and elastic modulus, which is consistent with the phenomenon in tensile simulations. At the end of elastic de-formation stage, the stress increases slowly as the configuration is strained, which is different from the stress-strain curves in tensile process.” In revised manuscript (page 13 line 427 - 437): “Uniaxial compression loading is carried out to investigate the evolution mechanism of microstructure for Al-2%Mg single crystal alloy under different temperatures. In Fig. 12, the atomic structure snapshots present the defects evolution at every 0.01% strain from the yield point under four temperatures condition (10 K, 100 K, 300 K and 500 K). Corresponding to the stress-strain curves in Fig. 11, the snapshots appear initial dislocation at the strain of the yield points. As the temperature increases, the strain of initial dislocation emission decreases obviously. The dislocations begin to grow and slide when the strain increases. Different from tensile processes, the proportion of HCP atoms decreases not obviously with the increase of the temperature at compression loading pattern. Moreover, more defective structures are generated at compression loading condition. In the process of compression simulation, the damage to the lattice structure is more intense.” In revised manuscript (page 15 line 491 – 495): “The effect of grain size on the plastic deformation of Al-2%Mg alloy is further studied at various grain sizes. The dislocation density of samples with five grain sizes (5.1 nm, 7.0 nm, 10.1 nm, 12.8 nm and 16.0 nm) is plotted in Fig. 14. The increase of grain size decreases the dislocation density at the same tensile strain. The reason is that the increase of grain size decreases the proportion of GB atoms.” "In revised manuscript (page 18 line 547 - 550): “From section 3.3.1, the plastic deformation mechanism depends on the grain size. The plastic deformation is dominated by GB movement in the small grain size, while it is dominated by dislocation slip in the large grain size.”

Round 3

Reviewer 1 Report

Thank you for answers and revision.

Author Response

Responses to the comments from the reviewers We greatly appreciate the reviewers and editor for their professional and constructive comments, all comments have important guiding significance for our work. According to the comments, we have carefully revised this manuscript. In the following, we have addressed the reviewer's comments point by point. It should be noted that the reviewer's comments are marked in black italic and the corresponding responses are in blue. All the corrections in the revised manuscript are highlighted in RED. Reply to the Reviewer #1: Thank you for answers and revision. Reply: Thanks.

Reviewer 2 Report

No real improvements were made to the paper. Most of the authors' replies and inserts to the text are unspecific and often irrelevant. I cannot recommend the paper for publication.